# ExPLAIND: Unifying Model, Data, and Training Attribution to Study Model Behavior

## Abstract

Post-hoc interpretability methods typically attribute a model's behavior to its components, data, or training trajectory in isolation. This leads to explanations that lack a unified view and may miss key interactions. While combining existing methods or applying them at different training stages offers broader insights, such approaches usually lack theoretical support. In this work, we present ExPLAIND, a unified framework that integrates all these perspectives. First, we generalize recent work on gradient path kernels, which reformulate models trained by gradient descent as a kernel machine, to realistic settings like AdamW. We empirically validate that a CNN and a Transformer are accurately replicated by this reformulation. Second, we derive novel parameter- and step-wise influence scores from the kernel feature maps. Their effectiveness for parameter pruning is comparable to existing methods, demonstrating their value for model component attribution. Finally, jointly interpreting model components and data over the training process, we leverage ExPLAIND to analyze a Transformer that exhibits Grokking. Our findings support previously proposed stages of Grokking, while refining the final phase as one of alignment of input embeddings and final layers around a representation pipeline learned after the memorization phase. Overall, ExPLAIND provides a theoretically grounded, unified framework to interpret model behavior and training dynamics.

## 1 Introduction

Understanding the latent mechanisms of deep neural networks remains one of the central challenges in machine learning (Rudin et al., 2021; Rai et al., 2024; Zhang et al., 2025). As models become increasingly complex, interpretability has become an increasingly crucial tool — not just for debugging or improving transparency, but for building trust, ensuring fairness, and guiding further development (Doshi-Velez & Kim, 2017). Much of the recent progress in interpretability has focused on attributing a model's behavior to one of three main factors: its components, the data it was trained on, or the dynamics of the training process itself.

However, these approaches are often applied in isolation. Explanations focused on model components may ignore the influence of individual training examples or how these components evolved during optimization. Data-centric explanations can overlook how different parts of the model internalize those examples. This fragmentation thus limits our understanding, leaving important interactions unexplored. While some work has probed training dynamics (Müller-Eberstein et al., 2023; Tigges et al., 2024; Prakash et al., 2024), their insights often remain only loosely connected to analyses of model architecture or input data and lack a theoretical connection between checkpoints.

**I. Model decomposes into influence scores**

$$\phi_s(\theta, x, x')_j$$

model components

training data

training

**II. Explain by accumulating scores from different perspectives and granularities**

$$\sum \phi_s(\theta, x, x')_j$$

Figure 1: The ExPLAIND framework is based on the decomposition of the model along its components, training data, and training steps. Explanations are obtained by accumulating the resulting influence scores.

To address this fragmentation, we propose **Ex**act **P**ath-**L**evel **A**ttribution **I**ntegrating **N**etwork and **D**ata (**ExPLAIND**), a unified framework that captures how data, model components, and training dynamics jointly influence model behavior. ExPLAIND builds on the Exact Path Kernel (EPK, Bell et al., 2023) view of gradient-based learning, but extends it to modern training regimes. With this, ExPLAIND provides a theoretically grounded lens through which we can analyze how individual training examples influence model components throughout training. This unified perspective helps, for example, to interpret emergent learning phenomena such as Grokking (Power et al., 2022).

Our work makes the following contributions:

(i) **Theoretical extension of EPK.** We generalize the EPK to modern training regimes with optimizers that include first- and second-order gradient estimates, weight decay, dynamic learning rates, and mini-batching (see Section 3). We empirically validate that our kernel accurately represents both a CNN on a vision task and a Transformer on a math task.

(ii) **ExPLAIND framework.** Based on this theoretical foundation, we derive novel influence scores that quantify how individual parameters, training samples and training steps contribute to model predictions (see Section 4). The framework can be applied at different levels of granularity and from different perspectives, such as parameter level, data level and training-step level, cf. Figure 1. We validate the effectiveness of these scores for model component attribution via competitive parameter pruning.

(iii) **Case study of Grokking.** To demonstrate the capabilities of ExPLAIND, we apply the framework to a Transformer model known to exhibit Grokking (see Section 5). For the widely studied modulo addition Transformer, we uncover a previously unreported alignment phase where input embeddings and final layers interestingly align around a representation pipeline learned in the preceding training steps.

Thereby, ExPLAIND offers a theoretically grounded, empirically validated, and practically useful framework for analyzing modern machine learning architectures in a holistic manner.

## 2 RELATED WORK

We highlight the most directly relevant works and provide a more detailed discussion of the related literature in Appendix B.

**Post-hoc interpretability.** Post-hoc interpretability methods typically attribute model behavior to one of *input features* (Ribeiro et al., 2016), *training data* (Koh & Liang, 2017), or *model components* (Alain & Bengio, 2018). However, many approaches lack a theoretical foundation (Lipton, 2017; Saphra & Wiegreffe, 2024; Doshi-Velez & Kim, 2017; Basu et al., 2020; Bae et al., 2022) and can trade faithfulness for plausibility of explanations (Jacovi & Goldberg, 2020). Other work has extended interpretability into the temporal dimension, attributing model behavior to the *training dynamics*. For example, probing or circuit finding has been applied at different model checkpoints to identify learning phases (Müller-Eberstein et al., 2023; Tigges et al., 2024; Prakash et al., 2024). However, these approaches typically treat each training step independently, lacking a theoretical framework for *integrating changes in model behavior over time*.

**Path kernel methods.** Gradient path kernels (Domingos, 2020) reformulate a model trained by gradient descent as a kernel machine. Bell et al. (2023) extended this perspective to an exact equivalence, deriving the *Exact Path Kernel* (EPK). However, their formulation does not cover realistic learning scenarios involving gradient updates based on first- and second-order estimates, weight decay, dynamic learning rates, and mini-batching. Central to the EPK reformulation is the stepwise comparison of training and test sample gradients via dot products. This connects the EPK to the Neural Tangent Kernel (Jacot et al., 2018), which it generalizes over the training trajectory. The data attribution method TracIn (Pruthi et al., 2020) also measures dot-product gradient similarities across training steps for data attribution, but it lacks a theoretical connection to model predictions.

## 3 AN EXACT PATH KERNEL EQUIVALENCE FOR ADAMW

ExPLAIND is based on the EPK by decomposing the model predictions into fine-grained units of influence along the training data, model parameters, and training steps.

To capture the dynamics of modern optimization, including weight decay, moment estimates, learning rate schedules, and mini-batching, we focus on the AdamW optimizer (Loshchilov & Hutter, 2019). Here, the parameter update at step $s \in \{1, ..., N\}$ is of the form

$$\theta_s = \widehat{\theta}_{s-1} - \alpha_s \cdot \frac{\widehat{m}_s}{(\sqrt{\widehat{v}_s} + \epsilon)} \quad \text{with} \quad \widehat{\theta}_{s-1} = \theta_{s-1} - \alpha_s \lambda \theta_{s-1} \tag{1}$$

where $\alpha_s \in \mathbb{R}_{>0}$ denotes the learning rate schedule, $\lambda \in \mathbb{R}_{>0}$ the weight decay, and $Batch_s \subseteq \mathcal{D}$ denotes the mini-batch drawn from the training set $\mathcal{D} = \{(x_1, y_1), \ldots, (x_M, y_M)\}$. The first and second moments $m_s, v_s$ and their bias-corrected estimates are computed recursively via

$$m_s = \beta_1 \cdot m_{s-1} + (1 - \beta_1) \cdot \nabla_\theta L(\theta_{s-1}) \qquad \widehat{m}_s = \frac{m_s}{1 - \beta_1^s}$$

$$v_s = \beta_2 \cdot v_{s-1} + (1 - \beta_2) \cdot (\nabla_\theta L(\theta_{s-1}))^2 \qquad \widehat{v}_s = \frac{v_s}{1 - \beta_2^s}.$$

To reformulate models trained with AdamW as kernel machines over their training trajectory, we generalize the EPK to account for weight-decay, mini-batching and moment estimates. The following theorem states our main result for AdamW. A corresponding corollary then specializes this extension to gradient descent with momentum and weight decay.

**Theorem 3.1** (Extension of Bell et al. (2023) to AdamW). *Let $f_\theta : \mathcal{X} \to \mathcal{Y}$ be a model with parameters $\theta \in \Theta$ mapping inputs $x \in \mathcal{X} \subseteq \mathbb{R}^I$ to outputs $y \in \mathcal{Y} \subseteq \mathbb{R}^O$. Further assume that the final parameters $\theta_N$ are the result of optimizing $f_{\theta_0}$ from an initialization $\theta_0$ on a dataset $\mathcal{D} = \{(x_1, y_1), ..., (x_M, y_M)\}$ with $M$ samples and loss $L : \mathcal{Y} \times \mathcal{Y} \to \mathbb{R}_{\geq 0}$ using AdamW with weight decay $\lambda \in \mathbb{R}_{\geq 0}$ over batches $Batch_s \subseteq \mathcal{D}$ and learning rates $\alpha_s \in \mathbb{R}_{>0}$, $s \in \{1, ..., N\}$. Then the final model prediction $f_{\theta_N}(x)$ of a sample $x \in \mathcal{X}$ decomposes into*

$$f_{\theta_N}(x) = f_{\theta_0}(x) - \sum_{k=1}^{M} \sum_{s=1}^{N} \phi_s^{test}(x) \cdot \phi_s^{train}(x_k)^\top \cdot \mathbf{a}_{k,s} - \sum_{s=1}^{N} \phi_s^{test}(x) \cdot \mathbf{r}_s \tag{2}$$

*where $\theta_s(t) := \theta_s - t(\theta_s - \theta_{s+1})$ is the linear mixture of parameters between step $s$ and $s+1$, and*

$$\mathbf{a}_{k,s} := \left( \frac{dL(f_{\theta_s(0)}(x_k), y_k)}{df_{\theta_s(0)}(x_k)} \right)^\top \in \mathbb{R}^O \qquad \phi_s^{test}(x) := \int_0^1 \nabla_\theta f_{\theta_s(t)}(x) \, dt \in \mathbb{R}^{O \times D}$$

$$\alpha_{s,i} := \alpha_s(1 - \beta_1)\beta_1^{s-i} \frac{\sqrt{1 - \beta_2^s}}{1 - \beta_1^s} \in \mathbb{R} \qquad \phi_s^{train}(x) := \sum_{i=0}^{s} \alpha_{s,i} \frac{\mathbf{1}_{x \in Batch_i} \nabla_\theta f_{\theta_i(0)}(x)}{\sqrt{v_s}} \in \mathbb{R}^{O \times D}.$$

$$\mathbf{r}_s := \alpha_s \lambda \theta_s(0) \in \mathbb{R}^D$$

*Sketch of Proof.* We follow similar arguments as Bell et al. (2023). The key difference is to start from the parameter update given in Eq. (1) and to account for mini-batches through indicator variables. The complete proof is in Appendix D.1. $\qquad \square$

**Corollary 3.2** (Gradient Descent with Momentum). *In the same setup as in Theorem 3.1, but for gradient descent with momentum $\beta$ and learning rates $\alpha_s$ with update equation*

$$\theta_s = \theta_{s-1} - \alpha_s \beta \mathbf{b}_s$$

*where $\mathbf{b}_s$ is defined recursively as*

$$\mathbf{b}_0 = \nabla_\theta f(\theta_0), \quad \mathbf{b}_s = \beta \mathbf{b}_{s-1} + \nabla_\theta f(\theta_{s-1}) + \lambda \theta_{s-1}$$

*we derive a similar decomposition as in Theorem 3.1, but with the regularization term changing to $\mathbf{r}_s = \alpha_s \sum_{j=0}^{s-1} \beta^{s-j} \theta_j(0)$.*

*Proof.* We follow a similar approach as in Theorem 3.1. The key difference is the absence of the second-order term and that the regularization term moves into the momentum term, which results in the reformulation of the respective terms. The full proof can be found in Appendix D.1. $\qquad \square$

We further derive the following corollary which decomposes the trajectories of intermediate model activations (such as layers) as well as the loss. We can use this later for the interpretation of intermediate model activations, for example, at an intermediate layer.

**Corollary 3.3** (Loss and intermediate activations). *Assume the setup of Theorem 3.1 holds. Furthermore, let the model be of the form $f_\theta = h_\kappa \circ g_\iota$. Then the loss $L(f_{\theta_N}(x), y)$ and the intermediate activations $g_\iota(x)$ decompose analogously to the equation in Theorem 3.1.*

*Proof.* For the loss, we can set $\tilde{L} = \text{id}$ and $\tilde{f}(x) = L(f_{\theta_N}(x), y)$ and apply Theorem 1. Analogously, we can set $\tilde{L}(y', y) = \tilde{L}(h_\kappa(y'), y)$ and $\tilde{f}(x) = g_\iota$ to derive the second statement. ☐

### 3.1 VALIDATION OF EXACT MODEL REPRESENTATION

To empirically validate the exact equivalence of Theorem 3.1 and Theorem 3.2, we compute the EPK for a Transformer model and a CNN.[1] For this, we evaluate different numbers of integration steps in the test feature map $\phi^{\text{test}}$ and summarize the results in Table 1. The derived formulation provides an accurate approximation for both models, reproducing $100\%$ of their classification decisions. Furthermore, the output distributions closely match those of the original models, indicated by near-zero KL divergences. We therefore use 100 integration steps for $\phi^{\text{test}}$ for the rest of this study.

Table 1: We report the accuracy of the EPK representation with respect to model predictions, and the output similarity measured by KL divergence. For 100 integration steps, the EPK matches the trained model.

| Model | ResNet9 | | Transformer | |
|---|---|---|---|---|
| Data | CIFAR-2 | | MOD-113 | |
| Integration Steps | 10 | 100 | 10 | 100 |
| EPK Accuracy | 0.997 | 1.0 | 0.748 | 1.0 |
| KL Divergence | 0.0 | 0.0 | 0.885 | 0.0 |

## 4 THE EXPLAIND FRAMEWORK

We now build on the established EPK of the training with realistic optimizers such as AdamW and define the *ExPLAIND* framework, which attributes prediction at the level of training samples, model parameters, and training steps. The key idea is a *tensor of influences* which indexes contributions by training steps, parameters, training samples, and outputs. Different explanations are then obtained by accumulating this tensor along selected axes, leading to parameter-, data-, and step-wise attributions at arbitrary levels of granularity, as visualized in Figure 1. We denote the influence of a parameter $\theta^{(i)}$ at step $s$ due to training sample $x_k$ on the prediction of class $j$ of a sample $x$ as

$$\psi_s(\theta^{(i)}, x, x_k)_j := \phi_s^{train}(x_k)_{j,i} \cdot \phi_s^{test}(x)_{j,i} \cdot (a_{k,s})_j \in \mathbb{R}$$

and the influence of the regularization at step $s$ on the same prediction as

$$\psi_s^{reg}(\theta^{(i)}, x)_j := \phi_s^{test}(x)_{j,i} \cdot (\mathbf{r}_s)_i \in \mathbb{R}.$$

We collect all atomic scores into one multi-dimensional object, with axes corresponding to *steps*, *parameters*, *test samples*, *training samples*, and *outputs*.

**Definition 4.1** (Tensor of Influences). *For a set of training steps $\mathcal{S} \subset \{1, 2, ..., N\}$, model parameters $\Theta \subset \{\theta^{(1)}, \theta^{(2)}..., \theta^{(D)}\}$, predictions $\mathcal{X}_{test} \subset \mathcal{D}_{test}$, training samples $\mathcal{X}_{train} \subset \mathcal{D}_{train}$, and outputs $\mathcal{J} \subset \{1, ..., O\}$ we define the respective **tensor of influences** as*

$$\Gamma_\mathcal{S}(\Theta, \mathcal{X}_{test}, \mathcal{X}_{train})_\mathcal{J} := (\psi_s(\theta, x, x')_j)_{s \in \mathcal{S}, \theta \in \Theta, x \in \mathcal{X}_{test}, x' \in \mathcal{X}_{train}, j \in \mathcal{J}} \tag{3}$$

*where, by convention, if $e$ is not already a set we identify it with the singleton $\{e\}$.*

Rewriting Theorem 3.1 using this definition, the $j$-th coordinate of the model output can be written as

$$f_{\theta_N}(x)_j = f_{\theta_0}(x)_j - \sum_{k=1}^{M} \sum_{s=1}^{N} \sum_{i=1}^{D} \psi_s(\theta^{(i)}, x, x_k)_j - \sum_{s=1}^{N} \psi_s^{reg}(\theta^{(i)}, x)_j. \tag{4}$$

Hence, the scores $\psi_s(\theta^{(i)}, x, x_k)_j$ are additive and sum up to the model prediction. The naturally arising backbone of our ExPLAIND framework is the accumulation of these scores over different (sub-)sets of parameters, training and test samples, as well as training steps. We make this explicit with the following definition.

---

[1]We provide our code in the supplementary material, and report technical details in Appendix C.

**Definition 4.2** (Accumulated influence). *Given the same setup as in Definition 4.1, the **accumulated influence** is the sum of the tensor of influences over selected axes:*

$$\Psi_{\mathcal{S}}(\Theta, \mathcal{X}_{test}, \mathcal{X}_{train})_{\mathcal{J}} := \text{sum}(\Gamma_{\mathcal{S}}(\Theta, \mathcal{X}_{test}, \mathcal{X}_{train})_{\mathcal{J}}) \tag{5}$$

*where* sum *sums up the elements along all dimensions of the tensor. Furthermore, we define leaving out a category in the arguments of $\Psi$ or $\Gamma$ as summing over the complete set along it.[2]*

*To account for differences in sign across influences to different predictions and outputs, we further define the **accumulated importance** as follows*

$$\bar{\Psi}_{\mathcal{S}}(\Theta, \mathcal{X}_{test}, \mathcal{X}_{train})_{\mathcal{J}} := \sum_{x \in \mathcal{X}_{test}} \sum_{j \in \mathcal{J}} |\Psi_{\mathcal{S}}(\Theta, x, \mathcal{X}_{train})_j|. \tag{6}$$

*We denote the analogous definitions for the regularization scores with $\Gamma^{reg}$, $\Psi^{reg}$, and $\bar{\Psi}^{reg}$.*

This formulation allows us to explain model behavior in a unified way across different dimensions. By choosing which axes of the influence tensor to keep and which to sum over, we obtain different *perspectives* such as parameter-level, data-level, or step-level attribution. Within each perspective, the *granularity* of the explanation depends on the size of the sets we consider. For example, if we want to study the influence of a single parameter $\theta^{(i)}$ on a given prediction of a sample $x$, we can accumulate parameter-wise kernel scores over the training set as $\Psi(\theta^{(i)}, x)$. In order to zoom out to the layer level per training step, we sum up the scores of all parameters that are a part of a layer $\Theta_L$ of the dataset at step $s$, i.e. compute $\Psi_s(\Theta_L, x)$. We provide more details and more examples in Appendix D.3 and illustrate the power of such explanations when studying Grokking in Section 5.

To investigate the relative influence of the regularization term, we furthermore define the accumulated parameter-wise difference of absolute influences.

**Definition 4.3** (Influence of the regularization). *With analogous definitions and conventions as above, we define the parameter-wise relative regularization influence on the model behavior as the difference*

$$D_{\mathcal{S}, \mathcal{X}_{train}, \mathcal{X}_{test}, \mathcal{J}}(\Theta) := \sum_{\theta \in \Theta} \left( \Psi_{\mathcal{S}}(\theta, \mathcal{X}_{test}, \mathcal{X}_{train})_{\mathcal{J}} - \Psi_{\mathcal{S}}^{reg}(\theta, \mathcal{X}_{test}, \mathcal{X}_{train})_{\mathcal{J}} \right). \tag{7}$$

Finally, we define the similarity of two predictions with respect to different parts of the model as the cosine similarity of their respective scores. This is a common avenue for interpreting high dimensional spaces, and allows us to compare the high dimensional tensors.

**Definition 4.4** (Similarity from the model's perspective). *With analogous definitions and conventions as above, we define the **similarity of two predictions** $x, x' \in \mathcal{X}$ as*

$$Sim_{\mathcal{S}, \Theta, \mathcal{X}_{train}, \mathcal{J}}(x, x') := \frac{\text{flat}(\Gamma_{\mathcal{S}}(\Theta, x, \mathcal{X}_{train})_{\mathcal{J}}) \cdot \text{flat}(\Gamma_{\mathcal{S}}(\Theta, x', \mathcal{X}_{train})_{\mathcal{J}})}{||\Gamma_{\mathcal{S}}(\Theta, x, \mathcal{X}_{train})_{\mathcal{J}}|| \cdot ||\Gamma_{\mathcal{S}}(\Theta, x', \mathcal{X}_{train})_{\mathcal{J}}||} \tag{8}$$

*where* flat *transforms a tensor into a vector and $|| \cdot ||$ is the vector $\ell_2$-norm.*

Similarly to the previous examples of influences of different granularities and unified perspectives, our notions of similarity and regularization influence can be generalized along the other dimensions, e.g., training samples, steps, or model components. Furthermore, we remark that explaining model behavior through the influence tensors $\Gamma$, the backbone of the ExPLAIND framework, can also be carried out using other forms of accumulation, which we leave to future work.

## 4.1 EFFICIENT IMPLEMENTATION

Recall that all previous derivations rely on accumulating slices of the influence tensors. Depending on the desired lens, this design choice has implications for the efficiency of the resulting computations. For example, for influence scores accumulated over the training data, the time-complexity is equivalent to a single training run. Materializing all scores up to parameter level is expensive, however, as the memory complexity alone is in $\mathcal{O}(NDMO)$ for $N$ training steps, $D$ parameters, $M$ training samples, and $O$ dimensions. By making a suitable choice restricting these dimensions, ExPLAIND gives an

---

[2]For example, $\Psi(\Theta, \mathcal{X}_{test}, \mathcal{X}_{train})$ is the influence summed over all steps $\mathcal{S}$ and all output dimensions $\mathcal{J}$.

interface to control the trade-off between high and low granularity of explanations, i.e. whether they are local or global along the aforementioned dimensions, and thereby also the level of faithfulness of the explanations, which comes at a higher cost as argued before.

Although our empirical study and demonstration of ExPLAIND in Section 5 is focused on small scenarios, this intuition offers a clear path for applying ExPLAIND to *larger architectures* and datasets. There, we propose to rely on (1) *(early) accumulation* of relevant parts of the influences to reduce computational and memory requirements (e.g. the gradients can be computed over the batches directly if the data-perspective is more coarse-grained), and (2) on *subsampling the influences* to reduce the number of steps, samples or parameters necessary to consider.

## 4.2 VALIDATION THROUGH PARAMETER PRUNING

To validate our EXPLAIND-derived parameter importance scores, we conduct pruning experiments on the CNN model, ranking each parameter $\theta$ by its kernel importance score $\Psi_S(\theta)$. For a fraction $c$ and number of parameters $D$, we only keep the TOP-$cD$ parameters and set the rest to zero. To retain the model's ability to predict, we do not prune the output layer. To contextualize our results, we compare them against an implementation of Li et al. (2017), a popular pruning method proposed to compress CNN models, which is based on an iterative procedure in which the model is

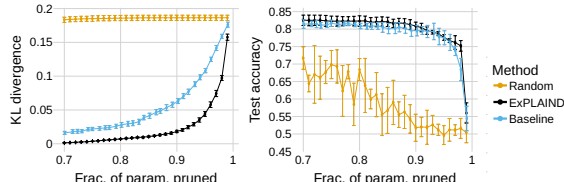

Figure 2: We prune the CNN weights and achieve test accuracy comparable to Li et al. (2017). Across all sparsity levels, the KL divergence of our model outputs is consistently lower. We report the means and standard deviations over 5 runs.

retrained in each step after pruning weights by magnitude importance. As shown in Figure 2, we find that our approach performs competitively against Li et al. (2017)'s approach on sparsity levels from 70% to 99%. Furthermore, our score-based, training-free pruning replicates the original model more closely than the baseline as is evidenced by the much lower KL divergences over all levels of sparsity. This underlines that our influence scores are indeed able to accurately quantify the influence of the parameters on the model's predictions. Note that this comparison is to validate that our method finds meaningful influence scores and does not aim to frame ExPLAIND as a SOTA pruning method.

## 5 GROKKING 'EXPLAIND'

Grokking is a training phenomenon where models first memorize, and then, after prolonged training, suddenly generalize (Power et al., 2022). Nanda et al. (2023) argue for multiple learning stages: (1) *memorization of the training data*, (2) *circuit formation*, in which the model learns a robust mechanism and thus generalizes, and (3) *cleanup*, during which the regularization "removes" memorization components. Other evidence emphasizes the role of dataset size, model capacity, and regularization (Huang et al., 2024; Wang et al., 2024a). We use ExPLAIND to revisit these hypotheses, integrating perspectives on model components, training data, and training dynamics in a single analysis. We further verify our insights through ablation experiments.

As a testbed, we use the well-studied modulo addition task (Varma et al., 2023) where the samples are of the form $[a][+][b][\mod 113 =]$ and the model predicts the correct result. We train a Transformer with AdamW (details in Appendix C.2). As shown in the top-left plot of Figure 3a, the model exhibits Grokking: training accuracy rises sharply, reaching nearly perfect performance around step 800. During this phase, test accuracy remains close to zero, but subsequently increases until reaching 100% at step 1939, at which point training is stopped.

### 5.1 INFLUENCE DECOMPOSITION

To study Grokking with ExPLAIND, we begin by decomposing the model's predictions into kernel and regularization influences over the full training trajectory (see Figure 3a). We then study their layer-wise decompositions. This provides a global view of the training dynamics before, during, and

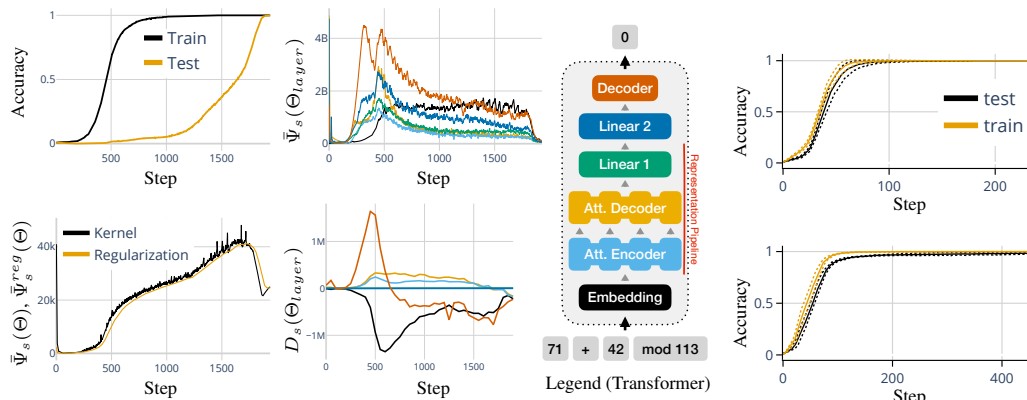

(a) **Left:** Accuracy and importances of kernel and regularization over the training of the Transformer model. **Right:** Layer-wise importance of the kernel (top) and influence of the regularization (bottom). As shown in the legend on the right, the Transformer consists of an attention layer (Att. Encoder and Att. Decoder) and an MLP (Linear 1 and Linear 2).

(b) We train a model initialized with the representation pipeline from the final model (**top**) and from step 1539 (**bottom**) and initialize the rest at random.

Figure 3: Training statistics, influences, and validation of our representation pipeline hypothesis.

after generalization, and highlights which model components drive these phases. ExPLAIND reveals the following trends in how kernel and regularization influences evolve across layers:

**Kernel versus regularization balance.** The absolute influences $\bar{\Psi}_s(\Theta)$ and $\bar{\Psi}_s^{reg}(\Theta)$ (bottom left of Figure 3a) grow together from the start of memorization (around epoch 200), both peaking near step 1700 before declining. In the final phase, $\bar{\Psi}_s^{reg}(\Theta)$ dominates which indicates that the final training phase is governed by a higher relative influence of regularization.

**Decoder dominance in memorization.** On the layer level, the memorization phase coincides with a peak in regularization influence $D_s(\Theta_{layer})$ (bottom right), and absolute influences $\bar{\Psi}_s(\Theta_{layer})$ (top right) of the decoder. This establishes the decoder as the most influential component for memorization.

**Middle-layer alternation and circuit formation.** The second peak of the decoder influence (top right) is preceded by peaks of the attention and linear layers, suggesting a single alternation between fitting the decoder and the intermediate layers. This marks the beginning of the circuit formation and hints at a representation pipeline forming in the middle layers.[3] Their influence then decreases, supporting this interpretation. With respect to regularization, both layers have a rather low influence, indicating that regularization acts primarily on the embedding and decoder layers.

**Late embedding dominance.** The embedding layer only begins to influence predictions in the circuit formation phase, evidenced by its sharp relative regularization dominance in $D_s(\Theta_{layer})$ and its rise in $\bar{\Psi}_s(\Theta_{layer})$. Remarkably, the latter continues to increase slightly during the rest of the training while all other layers' influences decrease during this phase.

These insights suggest that the model progresses through a sequence of shifts in influence: decoder-driven memorization, middle-layer alternation for circuit formation, and eventual embedding-decoder dominance under higher regularization influence. The rapid increase in test accuracy can thus be explained by a *simultaneous alignment of the input embeddings and decoder around the representation pipeline in the intermediate layers* (marked in red in the Transformer in Figure 3a).

To confirm that the alignment around this representation pipeline is causal, we train models initialized at random but replace only the Attention and Linear-1 layers with their grokked counterparts, which we hypothesize to be central to the representation pipeline. As shown in Figure 3b, this initialization leads to instant generalization within the first 200 training steps, and, in particular, bypasses the usual memorization phase. Additional experiments (Appendix Figure 8) confirm that even using only the attention layer produces the same phenomenon. Starting from earlier checkpoints leads to a similarly

---

[3]This interpretation is further supported by the simultaneous emergence of cyclic patterns on the influence level around steps 450 - 500 (see Section 5.2 and Appendix Figure 10).

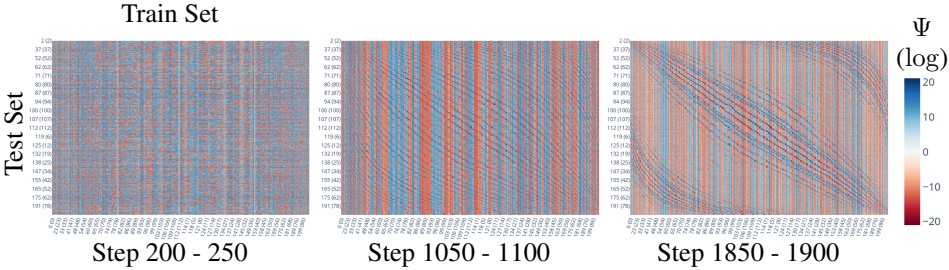

Figure 4: At different training stages, we show the influences $\Psi_S(\Theta_{dec}, x, x')$ of the training examples $x'$ on the test samples $x$ of the decoder layer $\Theta_{dec}$, summed over the output dimensions, for predictions on the test set and the training set, accumulated over the preceding 50 steps. The plots are labeled and ordered by the sum of inputs $a + b$ and the corresponding result $(a + b \mod 113)$. Corresponding figures for the other layers are provided in the Appendix in Figure 6.

rapid generalization, albeit with slightly lower final accuracy. This suggests a continuous refinement of the representation pipeline during the circuit formation phase.

To further probe this effect, we swap grokked layers into intermediate checkpoints (see Appendix Figure 8). The outer layers identified above consistently improve the predictions of these checkpoints, whereas other combinations decrease performance. This shows that the aligned final embedding and decoder layers can already operate effectively around a pipeline that is not yet fully generalized.

In sum, our findings refine the three-stage description of Grokking by Nanda et al. (2023) by showing that in the final phase the model outer' layers align around a generalized representation pipeline. They also connect to the efficiency perspective by Huang et al. (2024): our findings suggest that once a robust representation pipeline has formed, the influence of the regularization suppresses inefficient memorization solutions. In next section we show that this representation pipeline encodes a cyclic data geometry that becomes increasingly robust during training.

## 5.2 CYCLIC GEOMETRY

Next, we apply ExPLAIND from the data perspective to examine what structure the model learns. We study two complementary objects: $\Psi_s(\Theta_{dec}, x, x')$ (Figure 4) capturing how training samples $x'$ influence predictions $x$ with respect to the decoder and the *similarity matrices* $Sim_{\Theta_{layer}}(x, x')$ (Figure 5) which capture how samples are represented relative to each other. We find:

**Emergence of cyclic patterns in the kernel.** The vertical bands in Figure 4 reveal that at all training stages, each training sample has a global influence. After memorization, off-diagonal patterns emerge and sharpen, aligning with modular equivalence classes $(a + b \mod 113)$. Initially, these cycles have high frequency (about 2), i.e. the influence is strong for training samples whose label differences are a multiple of 2. Later in training, this cyclic geometry continuously shifts towards lower frequencies.

**Generalizable Data Geometry.** Similarity matrices (Figure 5) suggest a similar emergence of cyclic patterns, especially in the higher layers of the model. This indicates that the model indeed learns to represent the samples in a space where similarity is approximately a cosine of frequency 113. We test this hypothesis by fitting a Lasso regression to the similarity matrix accumulated over epochs 1850 to 1900 of the decoder, where we take as input features all cosines and sines of the pairwise differences of frequencies 2 to 113. The result is shown in Appendix Figure 9 and indeed confirms our qualitative observation with the coefficient of the cosine of frequency 113 being by far the largest. Furthermore, we note that in the final model this representation emerges only in the attention decoders, suggesting that the model first needs to combine the two input numbers and then proceeds to refining the cyclic geometry observed in higher layers.

Revisiting our layer-swapping experiment, when comparing the confusion matrices of the swapped and original checkpoints (see Appendix Figure 7 (b) to (e)), we observe that the systematic cyclic error patterns on the off-diagonals are greatly reduced once the final model's outer layers are swapped in, further supporting the cyclic geometry identified and suggesting that the alignment of the embedding and decoder indeed changes the prediction algorithm to one that uses the cyclic representations.

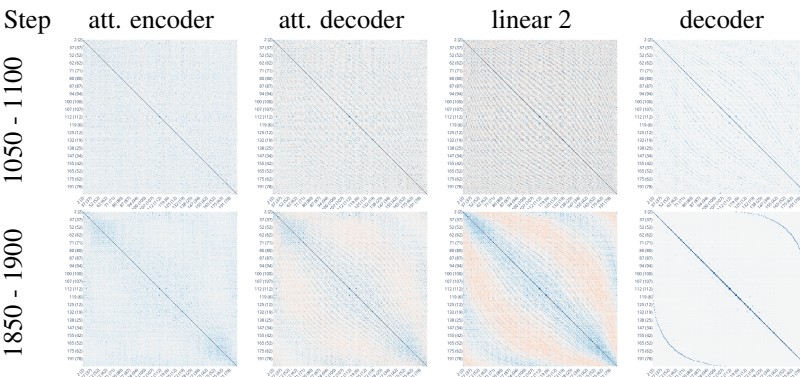

Figure 5: Similarity $Sim_{\Theta_{layer}}(x, x')$ of predictions of the test set of the Transformer model accumulated over different training stages. All layers and other training stages shown in Appendix Figure 10.

Our observations agree with Nanda et al. (2023), who find that neurons in the linear layers are computing combinations different cosines and sines. As ExPLAIND reveals, the fine-grained, neuron-level mechanisms that they describe result in a cyclic global pattern that is simple to interpret and the result of a continuous refinement from cycles of higher to lower frequency.

Taken together, these results show that Grokking reflects the progressive refinement of a cyclic data geometry and its alignment with input and output layers in the modulo addition Transformer.

# 6   DISCUSSION AND CONCLUSION

ExPLAIND offers a unified framework that bridges model components, data, and training dynamics, addressing a gap in post-hoc interpretability. Building on gradient path theory, it extends the Exact Path Kernel to realistic optimization regimes which is of independent interest. We validate the EPK representation and demonstrate the effectiveness of the resulting scores in parameter pruning. Through its theoretical foundation, ExPLAIND provides additive parameter-wise influence scores that can be aggregated to different levels of granularity and viewed from multiple perspectives. This positions ExPLAIND as a useful toolbox for unified attribution of model behavior.

Our exploratory study on Grokking highlights the utility of ExPLAIND by uncovering a novel perspective on its learning phases, with a central role for alignment and capability reuse. In particular, we identify an alignment phase characterized by the high relative influence of regularization on the outer layers preceded by the building of a representation pipeline. In grokked models, this suggests that further training serves to refine and re-use existing representations rather than build new ones. Once a robust latent structure has formed, generalization may emerge through alignment of input and output layers, challenging optimal training strategies.

**Limitations.** The ExPLAIND framework focuses on weight-level analysis and does not provide mechanistic or causal explanations. In addition, the insights by ExPLAIND are of qualitative nature and cannot yet be applied in an automated fashion. Besides, our study is limited to smaller models and tasks. Finally, computing our influence scores has high complexity (see Section 4.1), although we propose strategies to reduce runtime, in particular through increased granularity, which enable the analysis of larger scenarios.

**Future Work.** ExPLAIND should be used to study larger models through the lenses provided. In particular, it would be interesting whether our insights on Grokking in the modulo Transformer generalize to larger models and more complex tasks. More broadly, our results indicate that attributions to data and model components vary substantially across training, with critical patterns emerging at specific stages. Future interpretability methodology should therefore be designed to better surface these critical stages. Finally, more of possible perspectives and granularities provided by ExPLAIND should be studied. For example, Corollary 3.3 suggests that ExPLAIND can be used to interpret loss and activation level attributions, pointing to another promising direction.

REPRODUCIBILITY STATEMENT

The experimental methodology is described in detail in Appendix C, and all experiments are fully reproducible. Source code will be released upon acceptance and is also provided as part of the supplementary material. The proof of the main statement, Theorem 3.1, and Corollary 3.2 is included in Appendix D, the proofs of the remaining statements are part of the main text.

LLM USAGE STATEMENT

We used large language models (LLMs) for editing the manuscript, including for grammar, spelling, and rephrasing. We further use LLMs for support with coding. For both, we made sure to check the validity and security of all LLM outputs. AI tools do not contribute substantively to the ideas, research contributions, or results.

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

SUPPLEMENT TO THE PAPER "EXPLAIND: UNIFYING MODEL, DATA, AND TRAINING ATTRIBUTION TO STUDY MODEL BEHAVIOR"

## A  ETHICAL CONSIDERATIONS AND BROADER IMPACTS

**Interpretability for Fairness and Accountability.** Interpretability is a foundational requirement for building machine learning systems that are transparent, trustworthy, and legally accountable. Our framework, ExPLAIND, contributes to this goal by offering explanations that connect model behavior back to training data and model components. This is especially important in high-stakes domains (e.g. healthcare, criminal justice, finance), where decisions made by machine learning models must be auditable and understandable. Transparent systems are essential for identifying and mitigating biases, ensuring compliance with regulatory standards, and enabling meaningful human oversight.

**Causality, Overinterpretation, and Misleading Explanations.** Although ExPLAIND provides rigorous weight-level influence scores, they are inherently statistical and not causal. Misinterpreting these scores as direct causal claims about model behavior could lead to incorrect conclusions or misguided policy decisions. Practitioners and researchers must exercise caution when drawing inferences from post-hoc explanations and should clearly communicate the implications and limitations of an explanation.

**Respecting Data Ownership.** Recent investigations have revealed that major AI companies have utilized large-scale datasets containing pirated content, such as Library Genesis (LibGen), to train their models without obtaining permission from the original authors or rights holders. This practice not only infringes upon the intellectual property rights of creators but also raises significant ethical concerns regarding consent and fair compensation. Theoretically grounded attribution of training data and model components like ExPLAIND opens the door for mechanisms that acknowledge, attribute, and compensate the creators of influential data, thus respecting intellectual property rights.

**Unequal Access to Computational Resources.**

The development and application of computationally intensive interpretability methods, such as ExPLAIND, underscore a significant ethical concern: the disparity in access to necessary computational resources. This "compute divide" disproportionately favors well-funded industry players and elite academic institutions, enabling them to conduct advanced AI research and model auditing. In contrast, smaller institutions and independent researchers often lack the resources to engage in such work, limiting their participation in critical areas such as model interpretability and accountability. This imbalance not only hampers diverse contributions to the field but also raises concerns about whose models are scrutinized and whose voices are heard in shaping AI's future.

**Environmental Costs and the Role of Efficient Interpretability.**

Training large models, and by extension applying post-hoc interpretability methods like ExPLAIND, comes with significant computational and environmental costs. While our method is computationally expensive — often comparable to a single training run — we argue that this cost is justified in contexts where theoretical robustness and faithful attribution are necessary. Nonetheless, we acknowledge the environmental impact and advocate for minimizing computational overhead through algorithmic optimization, more efficient implementations and minimizing redundant applications. Future work should investigate scalable approximations of ExPLAIND to reduce emissions while preserving interpretability guarantees.

## B  EXTENDED LITERATURE REVIEW

This section provides an extended version of the literature review (see 2), including additional material relevant to the present work that was omitted from the main paper due to space constraints.

**Post-hoc interpretability.** There are many more approaches to post-hoc interpretability methodology that fall into one the three traditional explainability types, *input feature attribution* (Ribeiro et al., 2016; Lundberg & Lee, 2017; Binder et al., 2016; Zeiler & Fergus, 2013), the *training data attribution* (Park et al., 2023; Grosse et al., 2023; Chen et al., 2022; Ilyas et al., 2022; Bae et al., 2024; Liu et al., 2025; Ghorbani & Zou, 2019; Koh & Liang, 2017), and *model component attribution* (Tenney et al.,

2019; Wiegreffe & Pinter, 2019; Vig et al., 2020; Nanda, 2023; Arditi et al., 2024; Tang et al., 2024; Olah et al., 2020; Elhage et al., 2022; Rai et al., 2024).

**Grokking.** Grokking refers to a training phenomenon where models initially overfit but eventually generalize after prolonged training (Power et al., 2022). Liu et al. (2023) expanded this study to a broader suite of tasks and model architectures, providing a systematic characterization of Grokking's occurrence. More recent work (Wang et al., 2024b; Zhu et al., 2024; Huang et al., 2024) explores the implicit reasoning capabilities that arise during Grokking, the critical role of dataset size, and their connection to the double descent phenomenon. Nanda et al. (2023) argue that Grokking occurs in three phases: memorization, circuit formation, and cleanup. Our work refines this narrative, instead suggesting a progression through memorization, representation pipeline formation, and embedding-decoder alignment.

## C  TECHNICAL DETAILS AND HYPERPARAMETERS

In the next two sections, we specify the technical details of our models and data, as well as the hyperparameters we use. All implementation is provided in the supplementary material. In Section C.3 we detail the computation resources we used.

### C.1  CNN MODEL

We train a ResNet 9 model (He et al., 2016) with with 5 layers and two residual blocks, each consisting of two additional convolution layers with max-pooling, ReLU activations and a logarithmic softmax over the two dimensional output. We take the CIFAR-2 subset of CIFAR-10 (Krizhevsky et al., 2014) consisting of the classes dog and cat (10000 samples) and train using SGD with momentum of 0.9 for 12 epochs with a mini-batch size of 256 and weight decay of 0.005. We use a learning rate schedule that peaks in epoch 5 at 0.1. The loss is a cross entropy loss assuming logarithmic probability inputs.

### C.2  TRANSFORMER MODEL

The Transformer model, which was proposed by Varma et al. (2023) and used by Nanda et al. (2023), has a single layer encoder as described by Vaswani et al. (2017) and a *decoder* that consists of a single, fully connected layer mapping from the hidden dimension of 64 to the 115-token vocabulary. We use a 115-token input *embedding* without positional encoding, followed by a multi-head attention layer with four heads, each mapping to a space of dimension 16. We refer to the modules mapping to the lower dimensional spaces, that are used to compute the attention scores, as *attention encoder*, and accordingly call the modules reading from the representations after applying attention as *attention decoder*. The MLP layer on top of that consists of two fully connected layers (*Linear 1* and *Linear 2*), which map to and read from a 512-dimensional latent space. We visualize the transformer in the legend of Figure 3a.

We train on full batches using AdamW with a fixed learning rate of 0.001, weight decay of 4.0, and $\beta_1 = 0.98$, $\beta_2 = 0.99$ for the scaling parameters of the first and second moment estimates of the gradient, respectively.

The dataset consists of 4000 samples which each contain four tokens, namely the number [a], an addition token [+], the number [b] and the token [mod 113 =]. Here, $a, b \in \{0, 1, ..., 112\}$ and we always enforce $a \geq b$, leading to a total number of $\frac{113 \cdot 112}{2} = 6328$ possible data points of which we include 4000 randomly sampled ones in the train set and another 2000 in the test set labeled with the correct output token [c] containing the correct result $c = (a + b) \mod 113$ which has to be predicted.

### C.3  COMPUTE RESOURCES USED IN OUR EXPERIMENTS

Model training and retraining were carried out on a 20GB partition of NVIDIA A100 GPU for a total of less than 5 hours. Applying ExPLAIND to both models was much more compute intensive, resulting in about 20 hours of computation on a H200 GPU with 140GB GPU-RAM. Debugging and running the ablations presented, we carried out 12 such full runs of the EPK predictions computing ExPLAIND influence scores, leading to a total of about 240 H200 GPU-hours.

# D  PROOFS AND MATHEMATICAL DETAILS

## D.1  PROOF OF THE EPK EQUIVALENCE FOR ADAMW

We first restate the Theorem.

**Statement** (Kernel Equivalence for AdamW, repetition from Theorem 3.1 in main text)

*Let $f_\theta : \mathcal{X} \to \mathcal{Y}$ be a model with parameters $\theta \in \Theta$ mapping inputs $x \in \mathcal{X} \subseteq \mathbb{R}^I$ to outputs $y \in \mathcal{Y} \subseteq \mathbb{R}^O$. Further assume that the final parameters $\theta_N$ are the result of optimizing $f_{\theta_0}$ from an initialization $\theta_0$ on a dataset $\mathcal{D} = \{(x_1, y_1), ..., (x_M, y_M)\}$ with $M$ samples and loss $L : \mathcal{Y} \times \mathcal{Y} \to \mathbb{R}_{\geq 0}$ using AdamW with weight decay $\lambda \in \mathbb{R}_{\geq 0}$ over batches $Batch_s \subseteq \mathcal{D}$ and learning rates $\alpha_s \in \mathbb{R}_{>0}$, $s \in \{1, ..., N\}$. Then the final model prediction $f_{\theta_N}(x)$ of a sample $x \in \mathcal{X}$ decomposes into*

$$f_{\theta_N}(x) = f_{\theta_0}(x) - \sum_{k=1}^{M} \sum_{s=1}^{N} \phi_s^{test}(x) \cdot \phi_s^{train}(x_k)^\top \cdot \mathbf{a}_{k,s} - \sum_{s=1}^{N} \phi_s^{test}(x) \cdot \mathbf{r}_s \tag{9}$$

*where $\theta_s(t) := \theta_s - t(\theta_s - \theta_{s+1})$ is the linear mixture of parameters between step $s$ and $s+1$, and*

$$\mathbf{a}_{k,s} := \left( \frac{dL(f_{\theta_s(0)}(x_k), y_k)}{df_{\theta_s(0)}(x_k)} \right)^\top \in \mathbb{R}^O \qquad \phi_s^{test}(x) := \int_0^1 \nabla_\theta f_{\theta_s(t)}(x) \, dt \in \mathbb{R}^{O \times D}$$

$$\alpha_{s,i} := \alpha_s (1 - \beta_1) \beta_1^{s-i} \frac{\sqrt{1 - \beta_2^s}}{1 - \beta_1^s} \in \mathbb{R} \qquad \phi_s^{train}(x) := \sum_{i=0}^{s} \alpha_{s,i} \frac{\mathbf{1}_{x \in Batch_i} \nabla_\theta f_{\theta_i(0)}(x)}{\sqrt{v_s}} \in \mathbb{R}^{O \times D}.$$

$$\mathbf{r}_s := \alpha_s \lambda \theta_s(0) \in \mathbb{R}^D$$

*Proof.* Let $x \in X$ and $y_s = f_{\theta_s}(x)$. To rewrite each change $y_{s+1} - y_s$ in terms of a gradient flow, we parameterize the derivative of the parameters as follows

$$\frac{d\theta_s(t)}{dt} = \theta_{s+1} - \theta_s$$
$$\int \frac{d\theta_s(t)}{dt} dt = \int \theta_{s+1} - \theta_s \, dt \tag{10}$$
$$\theta_s(t) = \theta_s + t(\theta_{s+1} - \theta_s)$$

Each AdamW update step can be written as

$$\theta_s = \theta_{s-1} - \alpha_s \cdot \frac{\sqrt{1 - \beta_2^s}}{1 - \beta_1^s} \cdot \frac{m_s}{\sqrt{v_s}} - \alpha_s \lambda \theta_{s-1} -$$

Rewriting the Adam update rule we obtain

$$m_s = \beta_1 \cdot m_{s-1} + (1 - \beta_1) \cdot \nabla_\theta L(\theta_{s-1}) = \sum_{i=1}^{s} (1 - \beta_1) \cdot \beta_1^{s-i} \cdot \nabla_\theta L(\theta_{i-1})$$
$$\tag{11}$$
$$v_s = \beta_2 \cdot m_{s-1} + (1 - \beta_2) \cdot (\nabla_\theta L(\theta_{s-1}))^2 = \sum_{i=1}^{s} (1 - \beta_2) \cdot \beta_2^{s-i} \cdot (\nabla_\theta L(\theta_{i-1}))^2.$$

Note that the regularization term does not flow into the momentum term $m_s$ for AdamW. Combining the above, for the gradient flow, we can thus write

$$
\begin{aligned}
\frac{d\theta_s(t)}{dt} &= -\alpha_s \frac{\sqrt{1-\beta_2^s}}{1-\beta_1^s} \frac{m_s}{\sqrt{v_s}} - \alpha_s \lambda \theta_{s-1} \\
&= -\alpha_s \frac{\sqrt{1-\beta_2^s}}{1-\beta_1^s} \frac{\sum_{i=1}^{s}(1-\beta_1)\beta_1^{s-i}\nabla_\theta L(\theta_{i-1}(0))}{\sqrt{v_s}} - \alpha_s \lambda \theta_{s-1} \\
&= -\sum_{i=1}^{s} \alpha_{s,i} \nabla_\theta L(\theta_{i-1}(0)) \sqrt{v_s}^{-1} - \alpha_s \lambda \theta_s(0) \\
&= -\sum_{i=1}^{s} \alpha_{s,i} \cdot \left( \sum_{k=1}^{M} \frac{dL(f_{\theta_{i-1}(0)}(x_k), y_k)}{\partial \theta} \right) \sqrt{v_s}^{-1} - \alpha_s \lambda \theta_s(0) \\
&= -\sum_{k=1}^{M}\sum_{i=1}^{s} \alpha_{s,i} \cdot \frac{dL(f_{\theta_{i-1}(0)}(x_k), y_k)}{df_{\theta_{i-1}(0)}(x_k)} \frac{df_{\theta_{i-1}(0)}(x_k)}{\partial \theta} (\sqrt{v_s})^{-1} - \alpha_s \lambda \theta_s(0)
\end{aligned}
\tag{12}
$$

where we introduce the step learning rate to declutter the notation

$$
\alpha_{s,i} := \alpha_s(1-\beta_1)\beta_1^{s-i}\frac{\sqrt{1-\beta_2^s}}{1-\beta_1^s}.
\tag{13}
$$

Spelling out the dot product of the gradients via the sum that runs over index $j$, we can use this substitution to find that

$$
\begin{aligned}
\frac{df_{\theta_s(t)}}{dt} &= \frac{df}{d\theta} \cdot \frac{d\theta}{dt} \\
&= \frac{df}{d\theta} \cdot \left[ -\sum_{k=1}^{M}\sum_{i=1}^{s} \alpha_{s,i} \cdot \frac{dL(f_{\theta_{i-1}(0)}(x_k), y_k)}{df_{\theta_{i-1}(0)}(x_k)} \frac{df_{\theta_{i-1}(0)}(x_k)}{\partial \theta} (\sqrt{v_s})^{-1} - \alpha_s \lambda \theta_s(0) \right] \\
&= -\sum_{k=1}^{M}\sum_{j=1}^{D}\sum_{i=1}^{s} \alpha_{s,i} \cdot \frac{df_{\theta_s(t)}(x)}{\partial \theta_j} \frac{dL(f_{\theta_{i-1}(0)}(x_k), y_k)}{df_{\theta_{i-1}(0)}(x_k)} \frac{df_{\theta_{i-1}(0)}(x_k)}{\partial \theta_j} (\sqrt{v_s})_j^{-1} \\
&\quad - \alpha_s \lambda \frac{df}{d\theta} \theta_s(0)
\end{aligned}
\tag{14}
$$

where, for now, we stick to full-batch parameter updates in the substitution of the gradients and later account for the mini-batches through indicator variables. Since

$$
\frac{dL(f_{\theta_{i-1}(0)}(x_k), y_k)}{df_{\theta_{i-1}(0)}(x_k)} \frac{df_{\theta_{i-1}(0)}(x_k)}{\partial \theta_j} \in \mathbb{R},
$$

we have

$$
\frac{dL(f_{\theta_{i-1}(0)}(x_k), y_k)}{df_{\theta_{i-1}(0)}(x_k)} \frac{df_{\theta_{i-1}(0)}(x_k)}{\partial \theta_j} = \left( \frac{dL(f_{\theta_{i-1}(0)}(x_k), y_k)}{df_{\theta_{i-1}(0)}(x_k)} \frac{df_{\theta_{i-1}(0)}(x_k)}{\partial \theta_j} \right)^\top.
\tag{15}
$$

Combining that with the fact that for suitable matrices $A, B, C$ we have $A(BC)^\top = AC^\top B^\top$ we can rewrite

$$
\begin{aligned}
\frac{df_{\theta_s(t)}}{dt} &= -\sum_{k=1}^{M}\sum_{j=1}^{D}\sum_{i=1}^{s} \alpha_{s,i} \frac{df_{\theta_s(t)}(x)}{\partial \theta_j} \left( \frac{df_{\theta_{i-1}(0)}(x_k)}{\partial \theta_j}(\sqrt{v_s})_j^{-1} \right)^\top \left( \frac{dL(f_{\theta_{i-1}(0)}(x_k), y_k)}{df_{\theta_{i-1}(0)}(x_k)} \right)^\top \\
&\quad - \alpha_s \lambda \frac{df}{d\theta} \theta_s(0) \\
&= -\sum_{k=1}^{M}\sum_{i=1}^{s} \alpha_{s,i} \nabla_\theta f_{\theta_s(t)}(x) \left( \frac{\nabla_\theta f_{\theta_{i-1}(0)}(x_k)}{\sqrt{\widehat{v}_s}} \right)^\top \cdot \mathbf{a}_{i,k} - \alpha_s \lambda \nabla_\theta f_{\theta_s(t)}(x) \cdot \theta_s(0)
\end{aligned}
\tag{16}
$$

where we define

$$\mathbf{a}_{i,k} := \frac{dL(f_{\theta_{i-1}(0)}(x_k), y_k)}{df_{\theta_{i-1}(0)}(x_k)} = \left( \frac{\partial L}{\partial f_0} \ \frac{\partial L}{\partial f_2} \ \cdots \ \frac{\partial L}{\partial f_T} \right)^{\top}. \tag{17}$$

Using the second fundamental theorem of calculus, we can compute

$$\begin{aligned} f_{s+1}(x) - f_s(x) &= \int_0^1 \frac{df_{\theta_s(t)}(x)}{dt} dt \\ &= \int_0^1 \left( -\sum_{k=1}^{M} \sum_{i=1}^{s} \alpha_{s,i} \nabla_\theta f_{\theta_s(t)}(x) \left( \frac{\nabla_\theta f_{\theta_{i-1}(0)}(x_k)}{\sqrt{\widehat{v}_s}} \right)^{\top} \cdot \mathbf{a}_{i,k} - \right. \\ &\qquad \left. \alpha_s \lambda \nabla_\theta f_{\theta_s(t)}(x) \cdot \theta_s(0) \right) dt \\ &= -\sum_{k=1}^{M} \sum_{i=1}^{s} \alpha_{s,i} \left( \int_0^1 \nabla_\theta f_{\theta_s(t)}(x) \, dt \right) \left( \frac{\nabla_\theta f_{\theta_{i-1}(0)}(x_k)}{\sqrt{\widehat{v}_s}} \right)^{\top} \cdot \mathbf{a}_{i,k} \\ &\qquad - \sum_{s=1}^{N} \alpha_s \lambda \left( \int_0^1 \nabla_\theta f_{\theta_s(t)}(x) \, dt \right) \cdot \theta_s(0) \\ &= -\sum_{i=1}^{s} \sum_{k=1}^{M} \alpha_{s,i} \left( \int_0^1 \nabla_\theta f_{\theta_s(t)}(x) \, dt \right) \left( \frac{\nabla_\theta f_{\theta_{i-1}(0)}(x_k)}{\sqrt{\widehat{v}_s}} \right)^{\top} \cdot \mathbf{a}_{i,k} - \\ &\qquad \sum_{s=1}^{N} \alpha_s \lambda \left( \int_0^1 \nabla_\theta f_{\theta_s(t)}(x) \, dt \right) \cdot \theta_s(0). \end{aligned} \tag{18}$$

Combining over the full path of the gradients during training, we thus have

$$\begin{aligned} f_{\theta_N}(x) &= f_{\theta_0}(x) + \sum_{s=1}^{N} f_{s+1}(x) - f_s(x) \\ &= f_{\theta_0}(x) - \sum_{s=1}^{N} \sum_{i=1}^{s} \sum_{k=1}^{M} \alpha_{s,i} \left( \int_0^1 \nabla_\theta f_{\theta_s(t)}(x) \, dt \right) \left( \frac{\nabla_\theta f_{\theta_{i-1}(0)}(x_k)}{\sqrt{\widehat{v}_s}} \right)^{\top} \cdot \mathbf{a}_{i,k} \\ &\qquad - \sum_{s=1}^{N} \alpha_s \lambda \left( \int_0^1 \nabla_\theta f_{\theta_s(t)}(x) \, dt \right) \cdot \theta_s(0) \end{aligned} \tag{19}$$

which is of the form we wanted to derive. For mini-batch updates, we only need to consider the gradients $\nabla_\theta f_{\theta_{i-i}(0)}(x_k)$ of the training samples $x_k$ that were present in $Batch_i$. We express this by introducing an indicator variable $\mathbf{1}_{x_k \in Batch_i}$ to the train feature map which is one iff sample $x$ was in the mini-batch $Batch_i$ of step $i$. With this, we only calculate the train feature map with respect to the actual training samples involved in each step:

$$\begin{aligned} f_{\theta_N}(x) &= f_{\theta_0}(x) - \sum_{s=1}^{N} \sum_{i=1}^{s} \sum_{k=1}^{M} \alpha_{s,i} \left( \int_0^1 \nabla_\theta f_{\theta_s(t)}(x) \, dt \right) \left( \mathbf{1}_{x_k \in Batch_i} \frac{\nabla_\theta f_{\theta_{i-1}(0)}(x_k)}{\sqrt{\widehat{v}_s}} \right)^{\top} \cdot \mathbf{a}_{i,k} \\ &\qquad - \sum_{s=1}^{N} \alpha_s \lambda \left( \int_0^1 \nabla_\theta f_{\theta_s(t)}(x) \, dt \right) \cdot \theta_s(0) \\ &= f_{\theta_0}(x) - \sum_{k=1}^{M} \sum_{s=1}^{N} \phi_s^{test}(x) \cdot \phi_s^{train}(x') \cdot \mathbf{a}_k - \sum_{s=1}^{N} \phi_s^{test}(x) \cdot \mathbf{r}_s \end{aligned} \tag{20}$$

where we further define

$$
\phi_s^{test}(x) := \int_0^1 \nabla_\theta f_{\theta_s(t)}(x) \, dt
$$

$$
\phi_s^{train}(x) := \sum_{i=0}^{s-1} \alpha_{s,i} \frac{\mathbf{1}_{x_k \in Batch_i} \nabla_\theta f_{\theta_{i-1}(0)}(x_k)}{\sqrt{\widehat{v}_s}} \tag{21}
$$

$$
\mathbf{r}_s := \alpha_s \lambda \theta_s(0).
$$

$\square$

**Remark.** If the $\mathbf{a}_{i,k} = \mathbf{a}_k$ are constant over the training process, we can further simplify to

$$
f_{\theta_N}(x) = f_{\theta_0}(x) - \sum_{k=1}^M \left[ \sum_{s=1}^N \phi_s^{test}(x) \cdot \phi_s^{train}(x_k) \right] \cdot \mathbf{a}_k - \sum_{s=1}^N \phi_s^{test}(x) \cdot \mathbf{r}_s
$$

$$
= f_{\theta_0}(x) - \sum_{k=1}^M I(x, x_k) \cdot \mathbf{a}_k - \sum_{s=1}^N \phi_s^{test}(x) \cdot \mathbf{r}_s \tag{22}
$$

for function $I(x, x_k) := \sum_{s=1}^N \phi_s^{test}(x) \cdot \phi_s^{train}(x_k)$.

In practice, this holds for many scenarios. For example, it is the case for the logarithmic cross entropy loss we are using for our models. There, we assume the model outputs $f_{\theta_{i-1}(0)}(x_k)$ to be log-probabilities (implemented by a final log-softmax non-linearity) and thus have

$$
L(y_k, f_{\theta_{i-1}(0)}(x_k)) = -\sum_{c=1}^C (y_k)_c (f_{\theta_{i-1}(0)}(x_k))_c \tag{23}
$$

where $C$ is the number of classes and which implies that

$$
\mathbf{a}_{i,k} = \frac{dL(y_k, f_{\theta_{i-1}(0)}(x_k))}{df_{\theta_{i-1}(0)}(x_k)} = -y_k \tag{24}
$$

are constant over the training steps $i$.

**Remark.** If $f_{\theta_0}$ is constant, then the above reformulation is the regularized kernel machine (see Bell et al. (2023)). In order for $I$ to become a kernel function, one has to introduce a conditional, unified feature map that computes the correct result depending on the nature of the input sample.

### D.2 PROOF OF COROLLARY 3.2

For the CNN model, we use a GD optimizer where the regularization term introduced by weight decay flows into the momentum term. We therefore derive the EPK also for this optimizer.

We first restate the statement:

**Statement** (Kernel Equivalence for GD with Momentum, repetition of Corollary 3.2 in main text)

*We consider gradient descent with momentum $\beta$, learning rate schedule $\alpha_s$, and weight decay $\lambda$, i.e. the update equation is*

$$
\theta_s = \theta_{s-1} - \alpha_s \beta \mathbf{b}_s
$$

*where $\mathbf{b}_s$ is defined recursively as*

$$
\mathbf{b}_0 = \nabla_\theta f(\theta_0), \quad \mathbf{b}_s = \beta \mathbf{b}_{s-1} + \nabla_\theta f(\theta_{s-1}) + \lambda \theta_{s-1}.
$$

*We derive the same EPK decomposition*

$$
f_{\theta_N}(x) = f_{\theta_0}(x) - \sum_{k=1}^M \sum_{s=1}^N \phi_s^{train}(x_k) \cdot \phi_s^{test}(x)^\top \cdot \mathbf{a}_{k,s}^\top - \lambda \alpha_s \sum_{s=1}^N \phi_s^{test}(x) \cdot \mathbf{r}_s
$$

*where*

$$\mathbf{a}_{k,s} = \frac{dL(f_{\theta_s(0)}(x_k), y_k)}{df_{\theta_s(0)}(x_k)} \qquad \phi_s^{test}(x) = \int_0^1 \nabla_\theta f_{\theta_s(t)}(x)\, dt \qquad \mathbf{r}_s = \sum_{i=0}^{s-1} \alpha_{s,i}\theta_i.$$

$$\alpha_{s,i} = \alpha_s \cdot \beta^{s-i} \qquad \phi_s^{train}(x) = \mathbf{1}_{x \in Batch_i} \sum_{i=0}^{s-1} \alpha_{s,i}\nabla_\theta f_{\theta_i(0)}(x)$$

We note that compared to Theorem 3.1 the regularization term changes to

$$\mathbf{r}_s = \alpha_s \sum_{j=0}^{s-1} \beta^{s-j}\theta_j(0).$$

*Proof.* Writing out the recursive definition of $\mathbf{b}_s$, we thus have

$$\theta_s = \theta_{s-1} - \alpha_s\beta \left( \sum_{i=1}^{s} \beta^{s-i}\left(\nabla_\theta f(\theta_{i-1}) + \lambda\theta_{i-1}\right) \right)$$

$$= \theta_{s-1} - \sum_{i=0}^{s-1} \alpha_s\beta^{s-i}\nabla_\theta f(\theta_i) - \alpha_s \sum_{i=0}^{s-1} \lambda\beta^{s-i}\theta_i.$$

From this observation, we can follow analogous arguments as in the proof of Theorem 3.1 to obtain the EPK of GD with momentum. $\square$

### D.3 INFLUENCE ACCUMULATION

The ExPLAIND formulation of influence gives us a tensor that enables the attribution of model behavior to each of these dimensions across different, unified perspectives—these are the dimensions of influence we sum over—and granularities, corresponding to the size of the sets we sum over. Here we expand the examples given in the main text:

- **Single parameter.** The influence of a single parameter $\theta^{(i)}$ on a given prediction of a sample $x$, is given by the accumulation of parameter-wise kernel scores over the training set , i.e.

$$\Psi(\theta^{(i)}, x)_j = \sum_{k=1}^{M} \sum_{s=1}^{N} \sum_{i=1}^{D} \psi_s(\theta^{(i)}, x, x_k)_j.$$

- **Layer-level at a specific training step.** The influence of all parameters in a layer $\Theta_L$ at training step $s$ on the prediction for $x$ is

$$\Psi_s(\Theta_L, x) = \sum_{\theta^{(i)} \in \Theta_L} \sum_{k=1}^{M} \sum_{j=1}^{O} \psi_s(\theta^{(i)}, x, x_k)_j.$$

- **Data partition through a layer.** The influence of a layer $\Theta_L$ on $x$ due to a subset of training data $X \subseteq \mathcal{D}_{train}$ (for example, a data class) at step $s$ is

$$\Psi_s(\Theta_L, x, X) = \sum_{\theta^{(i)} \in \Theta_L} \sum_{x_k \in X} \sum_{j=1}^{O} \psi_s(\theta^{(i)}, x, x_k)_j.$$

# E   ADDITIONAL RESULTS

In this appendix, we provide additional figures from our experiments that were omitted from the main text.

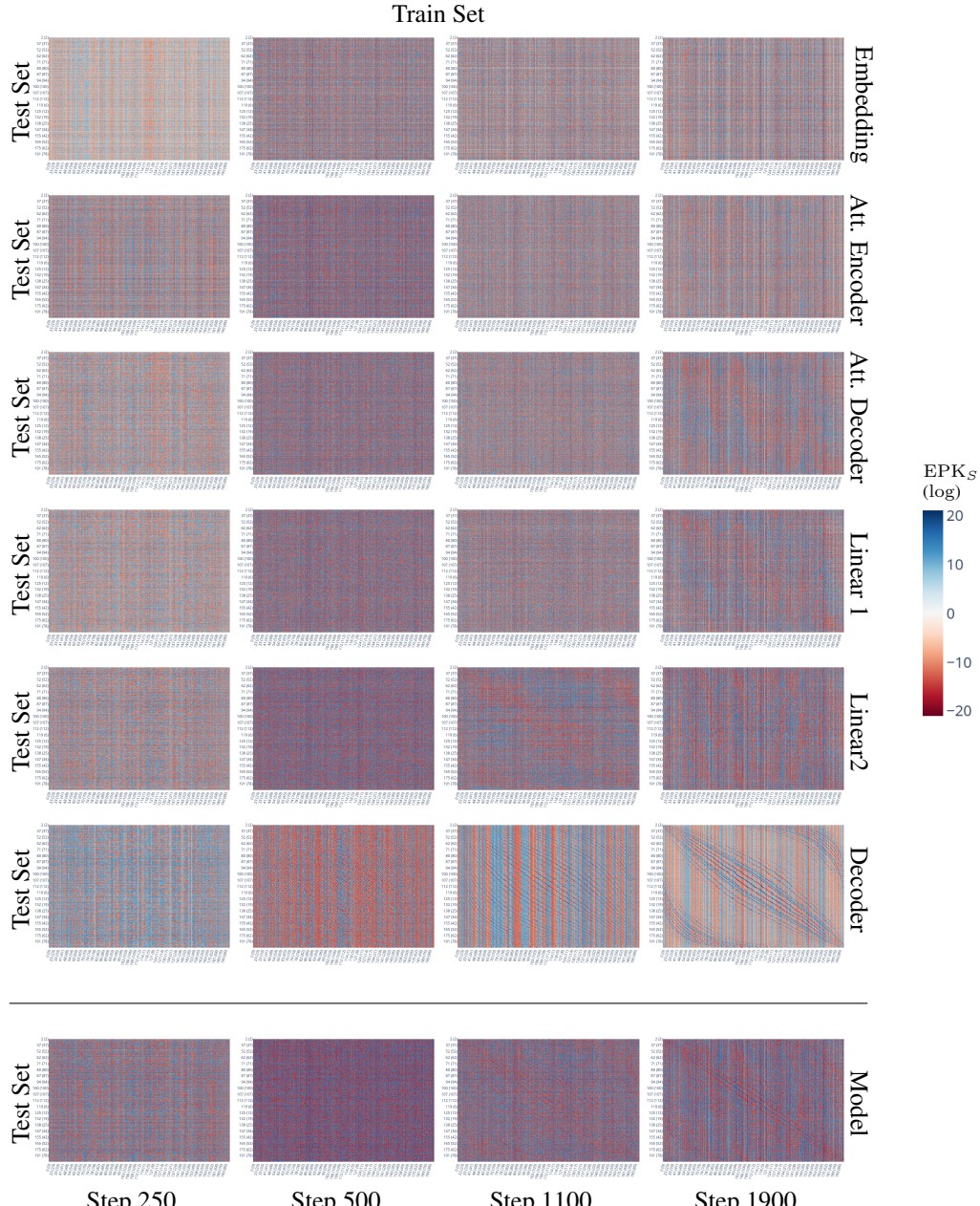

Figure 6: **Other slices of the kernel matrix of the Transformer model.** The EPK of all layers for predictions of the test set and the training set accumulated over preceding 50 steps, labeled with sum of inputs $a + b$ and respective result ($a + b \mod c$).

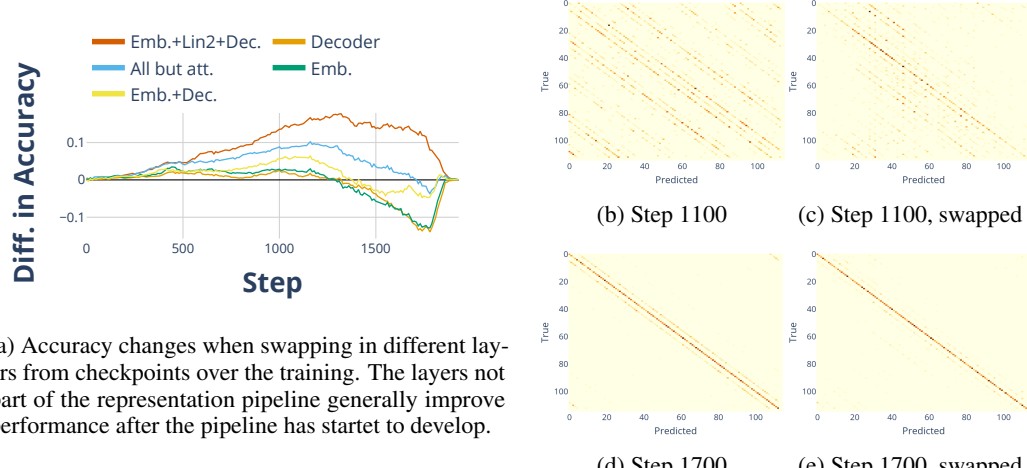

(a) Accuracy changes when swapping in different layers from checkpoints over the training. The layers not part of the representation pipeline generally improve performance after the pipeline has startet to develop.

(b) Step 1100

(c) Step 1100, swapped

(d) Step 1700

(e) Step 1700, swapped

Figure 7: **Layer swapping validations. Left:** We swap different layers of the final Transformer model into checkpoints across the training trajectory and find that the layers involved in the final alignment phase (the embedding, second linear layer and the decoder), improve accuracy by over $15\%$, supporting our hypothesis of a pipeline of intermediate layers developing a generalizing representation before the final Grokking phase. **Right:** Confusion matrices of two unedited checkpoints and their respective swapped versions. Note the decrease in systematic errors on the off-diagonals.

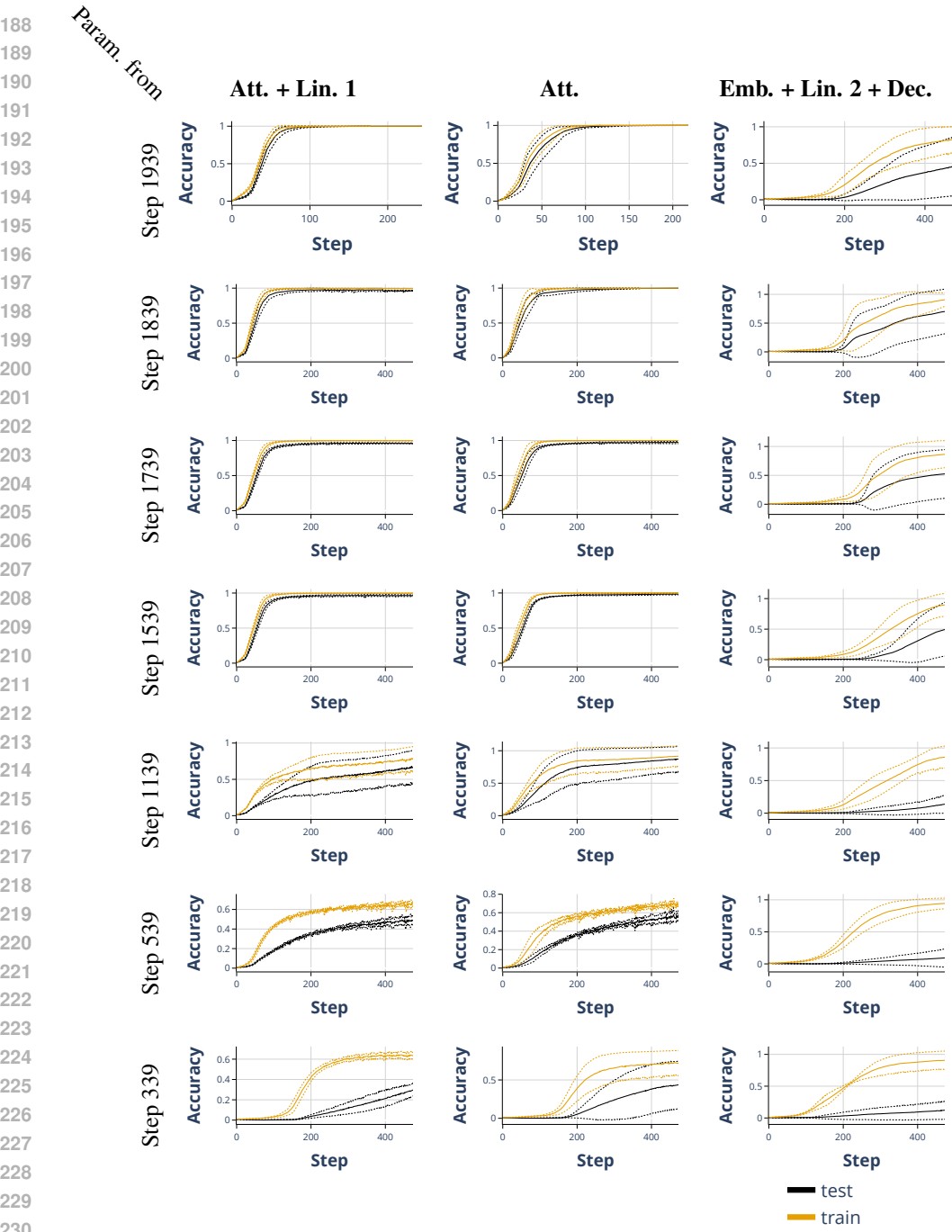

Figure 8: **Training on grokked intermediate representation pipelines.** We train a model initialized with the different parameters taken from different checkpoints and model components and initialize the rest at random. This leads to rapid, and direct generalization over 5 different runs when we take the attention weights (here 'Att.' refers to both the encoder and decoder of the attention layer) from later training steps, when the intermediate pipeline has already generalized. We report the mean over five runs and standard deviation as dotted error bars.

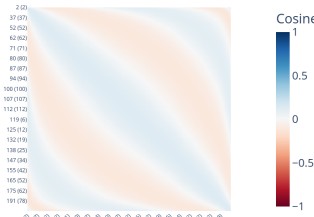

Figure 9: **Lasso regression on influence similarities.** We fit the second linear layers similarity domain introduced in steps 1850 to 1900 with a lasso regression. Features are the cosines and sines of frequencies from 2 to 113 of the pairwise differences of the sums of the samples. Shown: Predictions of the similarity as predicted by the regression. The resulting similarity pattern indicates that the model indeed learns to map the samples into space where distance is approximately a cosine of frequency 113. We report the exact regression coefficients in Table 2.

Table 2: **Lasso regression on influence similarities.** We fit the second linear layers similarity domain introduced in steps 1850 to 1900 with a lasso regression. Features are the cosines and sines of frequencies from 2 to 113 of the pairwise differences of the sums of the samples. The table shows all non-zero regressions coefficients of cosines frequency.

| cos frequency | Regression Coefficient |
| --- | --- |
| 113 | 0.112333 |
| 76 | 0.008232 |
| 51 | 0.004451 |
| 75 | 0.003616 |
| 52 | 0.001763 |
| 37 | 0.000679 |
| 13 | 0.000483 |
| 28 | 0.000259 |
| 38 | 0.000093 |

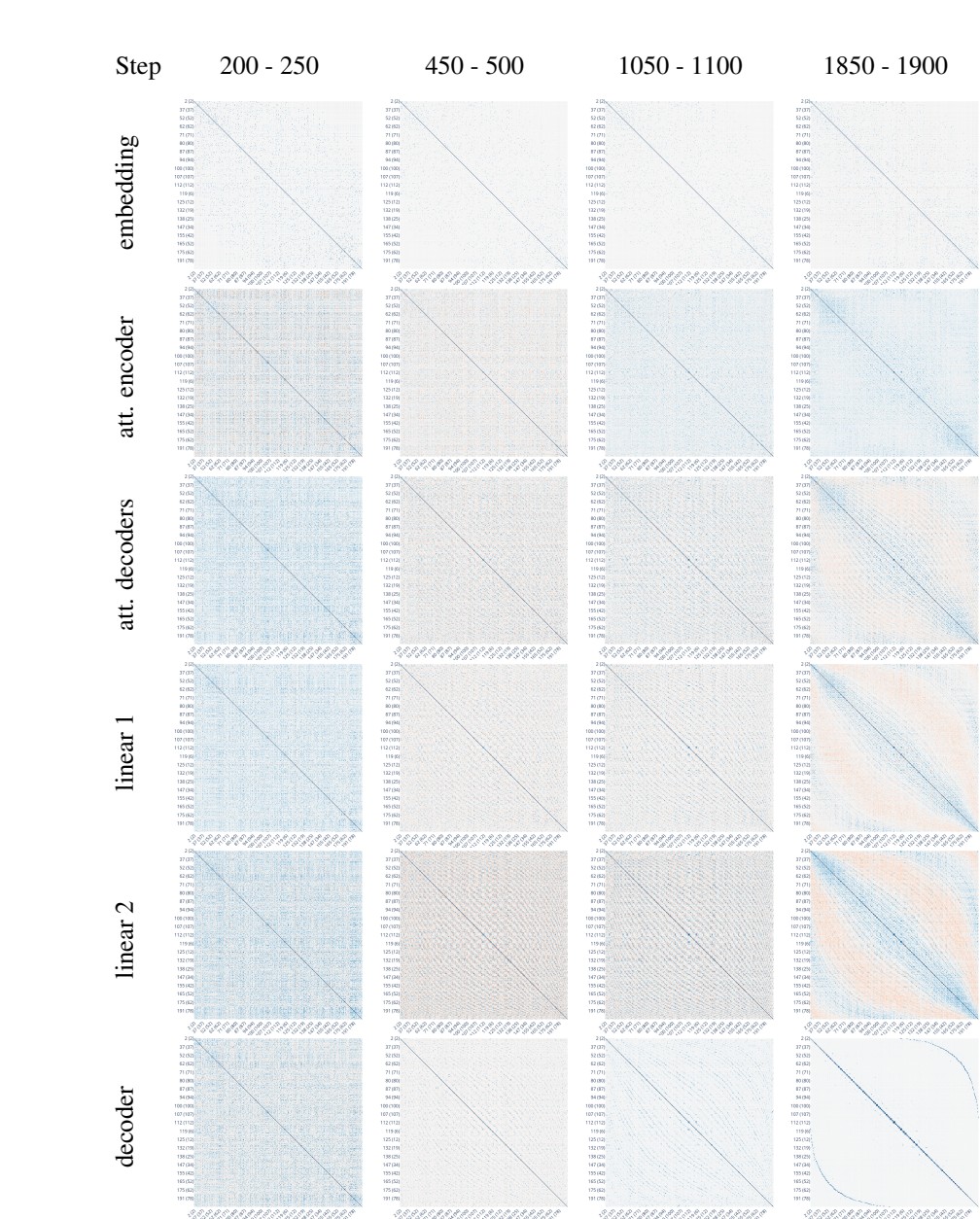

Figure 10: **Full similarity plots.** Similarity plots of test set predictions of the Transformer model accumulated over different training stages.