# OpenReview forum: "ExPLAIND: Unifying Model, Data, and Training Attribution to Study Model Behavior"
_ICLR.cc/2026/Conference — Submitted to ICLR 2026_

### Official Review · Reviewer_Sg8X · 2025-10-29

**Soundness:** 2
**Presentation:** 1
**Contribution:** 2
**Rating:** 4
**Confidence:** 3

**Summary:**

The paper proposes ExPLAIND, a unified framework for interpreting deep learning models by integrating model, data, and training step attribution perspectives. Building upon the EPK formulation, the paper extends previous results to modern AdamW, accounting for settings like weight decay, momentum, and mini-batching. From this theoretical foundation, the paper derives influence scores from different perspectives that are additive and can be accumulated along different dimensions. These scores are empirically validated through parameter pruning experiments and a case study on Grokking. Overall, the work proposes a theoretical unification of interpretability aspects.

**Strengths:**

- Theoretically extending EPK to practical AdamW optimizers with weight decay, momentum, and batching is a nontrivial generalization.

- The idea of unifying different attribution perspectives across model, data, and training step is commendable.

- The analysis of grokking is an interesting training dynamics analysis through attribution scores.

**Weaknesses:**

- Although the paper contains non-trivial theoretical derivations that seem to be sound, I am not fully convinced by its experimental evaluation. See questions below.

- The title of the paper can be a bit misleading and overclaiming. There are so many different interpretability works from all these three perspectives. The proposed framework seems to be one solution, but I don't see it unifying previous works. For example, model parameter attribution can include circuit discovery for IOI, linear probing directions for truthfulness, and universal neuron identification. How does the proposed score apply to these cases?

**Questions:**

- Although the paper claims a multidimensional tensor influence score spanning parameter, data, and training steps, only parameter-level pruning is evaluated in Section 4.2. Are there equivalent evaluations at the data level or training-step levels

- Also, the pruning experiment compares ExPLAIND only with one baseline? This single comparison does not provide strong evidence of superiority. It would be valuable to include other attribution-based baselines, similarly for other attribution levels.

- While the Grokking analysis is intriguing, several conclusions, like the dominance of regularization over kernel influences, or the significance of the peak at step 1700 in Figure 3(a), are not entirely clear to me. The curves for kernel and regularization appear roughly similar throughout, and the relationship between the accuracy curve and influence dynamics could be articulated more clearly, as 1700 doesn't see significant in the accuracy curve.

---

> ### Author Response · Authors · 2025-11-21
>
> Thank you very much for taking the time to read our paper and your thoughtful review. With regard to your criticisms and questions:
>
> > The title of the paper can be a bit misleading and overclaiming. There are so many different interpretability works from all these three perspectives. The proposed framework seems to be one solution, but I don't see it unifying previous works.
>
> We would like to emphasize that our goal is not to provide a meta-approach that unifies existing methodological approaches, but rather a framework that is able to provide explanations on the three perspectives (data, model components, and training) in a unified manner. It is not our intention to claim more than that and would be happy to change any parts of the paper that could be misleading in this regard, if you could point us to respective parts in the paper, thanks.
>
> Adding to the above, we would like to clarify that the ExPLAIND framework is not targeting to replace/outperform the approaches you mention. Rather, we offer a new paradigm whose explanations in the form of the accumulated influence scores integrate data, model components and training dynamics in a mathematically justified way. This makes it **a versatile, novel paradigm for interpretability which does not replace but add to the insights from approaches like circuit discovery, probing, or universal neuron identification.**
>
> > Are there equivalent evaluations at the data level or training-step levels
>
> Given the novelty of our unified paradigm to interpretability, we can indeed only compare along the non-unified dimensions of ExPLAIND.
>
> **For the training steps,** we are not aware of any standardized benchmarks. We think of our study on Grokking as a testbed for verifying ExPLAIND’s applicability to study training dynamics. There, we build hypotheses on the training (e.g. formation of a representation pipeline) which we test using the additional experiments.
>
> As you and the other reviewers highlight, **for data attribution,** there is indeed previous work attempting to benchmark such approaches. Since validating the data attribution dimension was concern across the other reviews, too, we decided to address this critique in the “**General Response on Data Attribution”** above.
>
> > Also, the pruning experiment compares ExPLAIND only with one baseline?
>
> We agree that a single baseline is not enough to establish ExPLAIND as a pruning method. We emphasize that the goal of the pruning experiment is to validate that the parameter influence scores are informative. Given the framework is not designed as a pruning method we believe the good pruning results is strong quantitative evidence of their explanatory value.
>
> > \[...\] the dominance of regularization over kernel influences, or the significance of the peak at step 1700 in Figure 3(a), are not entirely clear to me. The curves for kernel and regularization appear roughly similar throughout, and the relationship between the accuracy curve and influence dynamics could be articulated more clearly, as 1700 doesn't see significant in the accuracy curve.
>
> Thank you for raising this point. We agree that the explanation of these plots should be extended for clarity. In particular, we want to clarify that the accumulated curves in Figure 3 are a result of decomposing the model outputs. On the accuracy curve, changes in the model outputs might go unnoticed as predicting a correct result with more confidence has no effect on the curve. We will add this explanation to the Figure caption. Furthermore, we will plot the loss curve, whose trajectory follows the one of the output decomposition more intuitively (i.e. the sharpest drop in test loss happens at the peak at step 1700 you mention), in the same figure.
>
> **Please let us know, if these additional experiments and clarifications could address your concerns. We would be very happy to discuss further ways to improve the paper.**
>
> We noticed you rated our presentation score as 1, which stands in contrast to the other reviews. Would you have any concrete suggestions on how we could improve clarity and presentation?

---

### Official Review · Reviewer_ZFtG · 2025-11-05

**Soundness:** 3
**Presentation:** 3
**Contribution:** 3
**Rating:** 8
**Confidence:** 3

**Summary:**

This paper proposes ExPLAIND, a unified interpretability framework that decomposes a trained model’s prediction into additive influences from training data, model components, and training steps. Building on and extending the Exact Path Kernel view, the authors derive an exact decomposition for modern training regimes (notably AdamW) and demonstrate that the kernelized reformulation reproduces CNN and Transformer predictions.

**Strengths:**

- Solid theoretical contribution: Clear extension of the Exact Path Kernel to AdamW with mini-batching, moments, and weight decay, stated as a formal theorem.
- Empirical faithfulness check: The EPK reformulation matches original models’ decisions with 100 integration steps (accuracy 1.0; near-zero KL)
- Unified, additive attributions: Influence tensors can be summed along axes to obtain parameter/data/step views that directly tie to predictions.

**Weaknesses:**

-  Grokking analysis is insightful but mostly qualitative and on small models/tasks; generality to larger LLMs remains unproven.
- Pruning is used to “validate” importance scores rather than to deliver new SOTA compression results; comparisons are limited.

**Questions:**

None.

---

> ### Author Response · Authors · 2025-11-21
>
> Thank you very much for your encouraging feedback.
>
> We have introduced the following changes to the paper, which we hope will address some of your remaining concerns with our work:
>
> 1\. We have implemented an experiment showing that the training steps of a 100M parameter **GPT-2 model** trained on a factual reasoning task are also reconstructed by the ExPLAIND decomposition (using the tricks described in chapter 4.1). We will add the respective low reconstruction error to Table 1\. **This shows that ExPLAIND can be applied to larger scenarios**.
> As we write in our Future Work section (lines 480-481), we agree that the generality of our insights into Grokking/generalization should be verified in larger, more realistic settings. Our main goal is to introduce new methodology for this undertaking and we therefore consider such a comprehensive study out of scope for this paper.
>
> 2\. Furthermore, we will be more explicit with the motivation of the pruning experiment in the Limitations section to avoid confusion when it comes to the pruning results. We would like to emphasize that the pruning is performed as a validation of the importance scores. Due to its computational and memory overhead, ExPLAIND is designed as an interpretability framework, not a SOTA pruning method.
>
> Please let us know, if you have any additional questions or recommendations for improving your assessment of our work.

---

### Official Review · Reviewer_F7oz · 2025-11-06

**Soundness:** 2
**Presentation:** 2
**Contribution:** 2
**Rating:** 2
**Confidence:** 2

**Summary:**

The paper extends the Exact Path Kernel (EPK) to the case of optimization with weight decay. This stems from the historical observation that NN can be understood as kernel machines in a particular regime. This allows to define scalar quantifying the impact of a training exemplar on the prediction of a class, the *influence*.

These quantities are used to build a framework of interpretability. Then, the capabilities offered by this framework are benhcmarked on two toy tasks:
- parameter pruning of a CNN on Cifar-2
- the *grokking* of a small transformer on Mod 113

**Strengths:**

The theorem 3.1 is new, and brings the work of Bell et al closer to realistic experimental setups.

The sparsity gains in Sec 3.2 are interesting.

I love to see the "Emergence of cyclic patterns in the kernel" (l413). I wonder if we expect to see similar phenomenon on other family of problems (even artificial ones) that exhibit grokking?

Overall, the paper proposes an interesting line of ideas to understand training dynamics.

**Weaknesses:**

### Scope

Currently, I have issues with the motivation of the paper. While the extension of Bell et al to AdamW is interesting, I am less sure to understand the usefulness of the tool in general.

The experimental section is devoted to two setups: CNN training on Cifar-2 (cats and dogs) for sparsity, and the mod 113 task used to exhibit grokking. These two tasks are rather artificial. Even grokking as a whole received recent criticism in its ability to accurately describe some phenomena (see Jeffares and Shaar). I put an excerpt of their position here:

> This work argues that many prominent deep learning
phenomena discussed in the research literature are
not representative of challenges encountered in real-
world applications of deep learning. Thus, not all
efforts to understand these phenomena are equal in
value – we should focus on using them to refine our
broad explanatory theories of important aspects of
deep learning rather than developing narrow ad hoc
hypotheses to describe them in isolation. However,
this perspective is not consistently reflected in current
research practices within the community.

Jeffares, A. and van der Schaar, M., Position: Not All Explanations for Deep Learning Phenomena Are Equally Valuable. In Forty-second International Conference on Machine Learning Position Paper Track. 2025.

Overall, I struggle to grasp if the paper is selling a method ( in which case there is not enough evidence of usefulness), or simply analyzing grokking in Mod 113 with a specific toolbox (in which case it is overfitting a simple task with ad-hoc explanations).

### Clarity

I struggle to give a sense of quantities defined throughout the paper, like Tensor of Influences or Accumulated influence. The link with parameter pruning is not straightforward to me.  More toy examples could be useful, as sanity checks and for pedagogical purposes.

### Baselines

Better understanding of grokking in an artificial setup ring limited understanding of neural network training dynamic in broader settings. It could be interesting to apply these methods on bigger models trained on more realistic datasets.

Other tools try to explain networks' behavior from data points. We can mention influence functions, and Sobol' indices for variance decomposition. Since no comparison is made with existing tools, it is hard to situate the paper.

Mlodozeniec, B.K., Eschenhagen, R., Bae, J., Immer, A., Krueger, D. and Turner, R.E., Influence Functions for Scalable Data Attribution in Diffusion Models. In The Thirteenth International Conference on Learning Representations.

Fel, T., Cadène, R., Chalvidal, M., Cord, M., Vigouroux, D. and Serre, T., 2021. Look at the variance! efficient black-box explanations with sobol-based sensitivity analysis. Advances in neural information processing systems, 34, pp.26005-26014.

Same remark can be done for sparsity in Fig. 2. The baseline of Li et al (2017) is rather old.

**Questions:**

### Q1

In corollary 3.2, can you clarify if the regularization term is separated from the momentum (*à la* AdamW) or part of the loss?

### Q2

For mod 113, you rely on encoder/decoder architecture. For generations/question answering, decoders are typically sufficient. Why using an encoder/decoder pair here?

### Q3

In Section 5.1, what is your basis to label a phase "circuit formation"? For me, it looks like a (still unproven) hypothesis.

### Q4

Results in Fig 3.b are not very surprising. If my understanding of lines 370 is correct, it means that the last linear layer and projection onto vocabulary tokens are not in the final phase. This very much look like people typically do for finetuning of off-the-shelf foundation models: frozen  pretrained weights + finetuning of the head.  Can you comment on this?

---

> ### Author Response · Authors · 2025-11-21
>
> Thank you very much for taking the time to review our work. We appreciate your detailed feedback.
>
> With regards to your questions:
>
> **Q1:** Good catch\! As opposed to AdamW, the regularization term is not decoupled from the momentum term in this setup (as per the definition of standard momentum). Our implementations and proof account for this, but we will make this more explicit in the corollary to improve clarity.
>
> **Q2:** We follow the standard architecture from previous literature (see, e.g., \[1\]) for this. Indeed, this choice has been debated and it has been shown that Grokking can be shown for other architectures trained on this task. For more advanced tasks, we agree that other architectures might be more suitable.
>
> **Q3:** As we state in lines 307-308, we use terminology established in previous work \[1\].
>
> **Q4:** Our results indeed corroborate with earlier findings but our method provides additional insights beyond previous work. Through its influence and similarity over time, ExPLAIND provides a principled way to study and explain these phenomena. We agree that the connection to finetuning dynamics is interesting and will extend Future Work with this important idea.
>
> Regarding the weaknesses you raise:
>
> >  I am less sure to understand the usefulness of the tool in general
>
> Regarding your **uncertainties regarding the motivation of the paper,** we would like to reiterate that our main goal is to propose a new paradigm for interpretability, incorporating data, training, and model components (to which no comparable approach exists, which jointly accounts for all these ingredients). We selected our experiments to show the various ways ExPLAIND can be used to uncover the latent dynamics of model behavior. Given the large amount of possible use-cases of such an approach, we were not able to do this exhaustively in a single paper.
>
> More generally, we think the usefulness of an interpretability technique is measured by its ability to provide useful explanations besides its utility to do so. We believe that our case study on grokking illustrates that this is the case, i.e. through ExPLAIND we are able to provide useful explanations. We further validate those in our ablation experiments.
>
> >  Even grokking as a whole received recent criticism in its ability to accurately describe some phenomena (see Jeffares and Shaar).
>
> Thank you for raising this. We want to highlight that we do not make claims that would disagree with Jeffares and van der Schaar. We consider the Grokking model a suitable validation setup for ExPLAIND, as it is a widely used testbed for interpretability research (see, e.g., Nanda et al. \[1\] and related work). The task is small enough to exhibit rich, non-trivial training dynamics while remaining tractable for detailed analysis. Notably, grokking has served as one of the first examples of circuit discovery, an approach that has since become central in the interpretability of larger architectures. Moreover, its compact size allows for a comprehensive checkpoint-level analysis, whereas in the section on efficient implementation we demonstrate how ExPLAIND can be scaled to larger models.
> As we discuss in Future Work, we acknowledge that in order to make more general claims about Grokking, or generalization as a whole, a broader range of tasks need to be studied. Since our main goal is to introduce new methodology, we consider such a study out of scope for this paper.
>
> **Regarding larger models:** In Figure 1, **we will add a new experiment, namely, the step-wise reconstruction error of a GPT-2** (100M parameters)  model trained on a factual reasoning task (computed in less than 24 GPU hours). While the space restrictions of this paper prohibit us from a full analysis of this model, this shows that ExPLAIND can also be applied to large scale scenarios.
>
> **Data attribution validation** was also a concern of the other reviewers. Therefore, we decided to address this critique in the “**General Response on Data Attribution”** above.
>
> **As for the pruning experiments,** we would like to emphasize that the pruning is performed as a validation of the importance scores. Due to its computational and memory overhead, ExPLAIND is designed as an interpretability framework, not a SOTA pruning method. Therefore, even though we compare against an old baseline, we don’t think further baselines would add to this validation experiment.
>
> **Regarding clarity,** we will make sure to use the additional space of the final revision to extend the explanations of our definitions. In particular, we will extend on the motivation and intuition of the tensor of influences and accumulations. We also like your idea of introducing a small sanity check and will include in the form of an example discussing a linear model decomposed through ExPLAIND.
>
> Would that suffice to address your concerns? Please let us know if you have any further questions or what other improvements would help the quality of the paper.

---

> > ### Author Response · Authors · 2025-11-21
> >
> > **References:**
> >
> > \[1\] Nanda, Neel, Lawrence Chan, Tom Lieberum, Jess Smith, and Jacob Steinhardt. “Progress Measures for Grokking via Mechanistic Interpretability.” arXiv:2301.05217. Preprint, arXiv, October 19, 2023\. [https://doi.org/10.48550/arXiv.2301.05217](https://doi.org/10.48550/arXiv.2301.05217).

---

### Official Review · Reviewer_6LT3 · 2025-11-06

**Soundness:** 2
**Presentation:** 3
**Contribution:** 1
**Rating:** 2
**Confidence:** 4

**Summary:**

ExPLAIND introduces a unified framework for attributing neural network predictions to model components, training data, and training steps simultaneously, addressing the fragmentation of existing post-hoc interpretability methods that examine these factors in isolation.

 The framework extends the Exact Path Kernel formulation to realistic training scenarios by generalizing it to AdamW and momentum SGD optimizers with weight decay, learning rate schedules, and mini-batch updates through Theorem 3.1 and Corollary 3.2. The authors validate exactness empirically on a ResNet-9 trained on CIFAR-2 (binary classification, 10,000 samples) and a single-layer Transformer on modular addition (mod-113, 4,000 training samples), achieving perfect decision agreement and near-zero KL divergence with 100 integration steps.

The core contribution is a tensor of influences indexed by training steps, parameters, samples, and outputs, enabling multi-granularity attribution through flexible accumulation along different axes. Parameter-level scores are validated via competitive pruning experiments against magnitude-based baselines.

 A detailed grokking case study reveals decoder-driven memorization, middle-layer pipeline formation, and a late alignment phase where embeddings and decoder synchronize around learned representations, supported by layer-swapping ablations and cyclic geometry analyses in influence space.

However, the work is limited to small models, requires storing full training trajectories, scales as O(NDMO) in memory, and provides no principled granularity-selection guidance.

**Strengths:**

1. Rigorous theoretical extension: Theorem 3.1 extends EPK from basic gradient descent to AdamW with realistic training dynamics (weight decay, first/second moment estimates, mini-batching, learning rate schedules). The mathematical derivation is sound with complete proofs in Appendix D.1.
2. Exact model representation: Unlike approximate methods, ExPLAIND achieves perfect equivalence with the original model (100% accuracy match, zero KL divergence in Table 1) when using sufficient integration steps.
3. Unified mathematical framework: Successfully integrates parameter-level, data-level, and step-level attribution into a single tensor of influences, providing a principled mathematical object for multi-perspective analysis.

4.Insightful Grokking analysis: The Grokking case study shows that ExPLAIND can yield interpretable, insights into model training dynamics.

**Weaknesses:**

1. Gap between theoretical contribution and practical utility and scalability limitations:

     The EPK extension (Theorem 3.1) is mathematically sound and interesting.
     But the paper claims to provide a practical interpretability framework.
     Demonstrated only on toy problems with manual analysis and no path to scale (ResNet9 on 2-class CIFAR subset, small Transformer     on algorithmic task).

     O(NDMO) memory complexity: N steps × D parameters × M samples × O outputs.

     Computational cost: ~240 H200 GPU-hours for toy experiments suggests prohibitive costs for realistic models.


2. ExPLAIND requires training trajectory information (checkpoints $\theta_s$, gradients ($\nabla_{\theta} f_{\theta_s}(x_k)$)for all samples, optimizer states $m_s$ and $v_s$, batch membership indicators) at every training step. That questions its applicability to pretrained model where only final weights are available.


3. No principled methodology for aggregation selection: Framework provides 5-dimensional influence tensor but missing guidance on which aggregations are meaningful for which questions. Also, why Grokking analysis uses layer-level aggregation?

4. Insufficient validation and missing baselines: No comparison with other modern influence methods.

5. Grokking insights purely qualitative: No statistical testing of identified phases, no quantitative metrics, no automated discovery validation.

6. Parameter pruning only validates parameter scores, not the data attribution claims central to the paper.

7. Integration steps hyperparameter: Table 1 shows 10 vs 100 integration steps for $\phi^{test}$. They use 100, but no analysis of sensitivity or guidance on choosing this for new problems.
8. Grokking generalization unclear: Does the alignment phase insight generalize beyond modular arithmetic? The phenomenon might be task-specific, limiting broader impact of the case study.
9. Unclear justification for “influence” terminology:
The method defines influence scores purely via additive decomposition of the model output.
However, additivity alone does not imply causal or functional influence. Without sensitivity or perturbation tests, calling these quantities “influence” may be misleading.

**Questions:**

1. How can ExPLAIND be applied to pretrained models such as BERT, ViT, or deeper variants of ResNet when only the final model weights are available? Additionally, how computationally intensive would such an application be?

2. Aggregation methodology: What is your principled method for choosing:
     Which axes of the influence tensor to aggregate over?
     What granularity (parameter-level, layer-level, etc.)?
     For the Grokking analysis, why layer-level? Did you try other granularities and find similar patterns?
3. Mini-batch identification: When 256 samples are in $Batch_i$  and you compute $\nabla_{\theta} L$ on the full batch, how do you identify individual sample contributions when decomposing via $1{x_k ∈ Batch_i}$? The gradient is computed once for the batch, not separately per sample.
4. Comparison to Influence estimation methods: Standard influence estimation methods work on any pretrained model and provide causal approximations. In what scenarios would a practitioner choose ExPLAIND (requiring full training trajectory logging) over influence functions?
5. Cyclic geometry emergence: Figure 5 and 10 show interesting cyclic patterns. Do these emerge in other modular arithmetic tasks (division, multiplication)? Are they specific to the mod-113 Transformer architecture?
6. Have you tested whether removing parameters or samples with high ExPLAIND influence scores changes model predictions proportionally (validating the notion of “influence”)?

7. Is there a way to subsample or compress the influence tensor without losing theoretical correctness?

---

> ### Author Response · Authors · 2025-11-21
>
> Thank you very much for your review and questions! With regard to your questions and criticisms, we would like to answer as follows:
>
> **Model scale (W1 and Q1):** ExPLAIND can also be applied to larger scenarios using the strategies we outline in Section 4.1. No empirical evidence has been provided for this yet, which is **why we have computed a decomposition of the loss of a 100M parameter GPT-2** model trained on a factual reasoning task. There, we use the same tricks outlined (decomposition over partitions of the data and a subsample of steps) and find that each step is reconstructed accurately. We will add the respective low error to the results in Table 1 and report the computational cost (less than a day on an A100 GPU). Using the same strategies, the other models you mention would result in similar compute times.
>
> **Reliance on model checkpoints (W2):** Indeed ExPLAIND requires access to model checkpoints, which hinders the analysis of many SOTA models that do not publish them. This, however, is true for all approaches targeting training dynamics. For closed source models, ExPLAIND could at least be used by the developers internally, if they require such a unified interpretability perspective. In addition, more and more open models now also publish intermediate checkpoints enabling such analysis.
>
> **Aggregation selection (W3 and Q2):** Many interpretability techniques are rather qualitative than quantitative. Therefore, the level of aggregation you choose depends on the specific question you want to answer. We found that reducing the representation to two or three axes is most intuitive for us humans: For example, using a 1D quantity over time (such as the influence per parameter group, where we chose layers as a natural division of the architecture) or a 2D quantity over time (as in the case of kernel scores). Higher-dimensional explanations are generally harder to interpret as a starting point, but they can still be valuable for addressing specific, well-defined questions. Here, we want to position ExPLAIND as a versatile tool with more flexibility than other approaches (see Section 1&2).
>
> **Mini-batch identification (Q3):**  Yes, during the backward pass indeed the sample-wise gradients are aggregated per default, this is why, during the ExPLAIND computation, we recompute the sample-wise gradients (as long as you want to do the influence sample wise, i.e. retrieve the influence of a sample within a batch). We don't store gradients but recompute them if needed along the respective batch.
>
> **Data Attribution (W4, W6, Q4):** As data attribution validation was also a concern of the other reviewers, we address this critique in the **General Response on Data Attribution** above.
>
> **Insights into Grokking (W5):** We would like to clarify that our scores of influence in itself are exact atoms of the decomposed model quantity (i.e. the outputs, loss, etc.), i.e. a quantitative not qualitative way to investigate model behaviour (see Theorem 3.1). As is common in interpretability approaches, we offer different ways to derive human-interpretable explanations from these scores (e.g. layer-wise influence over time, similarity matrices, influence of dataset on predictions, etc.). While we agree this lacks statistical guarantees and requires user effort, this approach is established in the field of interpretability and should not be grounds for rejection (see, e.g., probing \[1\] or circuit analysis \[2\]). In order to make this more transparent in the paper, we will **extend the limitations section to include this.**
>
> **Integration steps hyperparameter (W7):** We ablate the number of integration steps for 10 and 100 steps and find that, while 10 steps might be enough for many scenarios (e.g. the CNN model), 100 steps yields a more accurate decomposition (but result in longer runtimes). These numbers agree with the findings of Bell et al. (2023), and thus should provide a helpful heuristic for future applications of ExPLAIND.
>
> **Grokking generalization unclear (Q5 and W8):** Further study of Grokking through ExPLAIND is a promising future direction. We agree that the phenomenon we observe is specific to the modular arithmetic setup and its generalization to other tasks is unclear. We discuss this in our Future Work section (lines 480-481). However the purpose of the current submission is mainly to introduce and establish the ExPLAIND framework through theoretical derivations, validation experiments and an applicability study. Therefore, we think that an extensive analysis across more tasks and models is out of scope for this paper.

---

> > ### Author Response · Authors · 2025-11-21
> >
> > **Unclear justification for “influence” terminology (Q6 and W9)**: Our framework provides an exact decomposition of the model outputs. Our motivation to call the atoms of this decomposition as 'influence scores' is thus strictly mathematical as established in Theorem 3.1. The additivity of these scores is mathematically implied by this theorem and not an assumption or engineering trick on our side.
> >
> > **Subsampling or compressing the scores (Q7):** As we describe in Section 4.1, ExPLAIND scales by limiting the scope of its explanations to, e.g., single steps, data partitions, and model layers. While this approach will not lead to scores that reconstruct the entire training, the decomposition per step is still accurate as is shown in the new experiment we introduce decomposing GPT-2 using this strategy.
> >
> > We hope this addresses your concerns regarding our work. Please let us know if you have any further questions or what other improvements would help the quality of the paper.
> >
> > **References:**
> >
> > \[1\] Alain, Guillaume, and Yoshua Bengio. “Understanding Intermediate Layers Using Linear Classifier Probes.” arXiv:1610.01644. Preprint, arXiv, November 22, 2018\. [https://doi.org/10.48550/arXiv.1610.01644](https://doi.org/10.48550/arXiv.1610.01644).
> >
> > \[2\] Olah, Chris, Nick Cammarata, Ludwig Schubert, Gabriel Goh, Michael Petrov, and Shan Carter. “Zoom In: An Introduction to Circuits.” *Distill* 5, no. 3 (2020): e00024.001. [https://doi.org/10.23915/distill.00024.001](https://doi.org/10.23915/distill.00024.001).

---

### Author Response · Authors · 2025-11-21
**General Response on Data Attribution**

We thank the reviewers for their valuable feedback. Reviewers 6LT3, F7oz, and Sg8X all comment on a lack of validation regarding the data attribution perspective of ExPLAIND. Adding further experiment results, clarifications, and discussion, we want to address their concerns as follows:

**Comparison to other data influence estimation methods:** Influence functions (IF, \[4\]) estimate the influence of the data under the leave-one-out-and-retrain paradigm \[1\], i.e. they answer the question “if I retrain my model without this sample, how much would my predictions change?”. Evaluating within this paradigm, we implemented an additional experiment on the CNN model using the linear data modeling score (LDS; \[2\]), and find that ExPLAIND (LDS=0.065) performs comparable to the popular baselines TRAK (LDS=0.055) and TracIn (LDS=0.055).  We will add this new experiment to the paper.

Furthermore, we want to clarify that **the IF-style task is different from the kind of data attribution ExPLAIND has to offer:** Our influence scores are the mathematically exact influence decomposition of each sample on the given model instance and prediction (as opposed to retraining the model with left out data). In addition, previous work targeting the IF objective does not consider training dynamics or model components. In other words, with respect to training dynamics interpretability, ExPLAIND allows to answer questions of influence shift over the training and model components, while other approaches don’t. We agree that we haven’t made these arguments explicit enough. We will therefore **include a discussion on data attribution** with the arguments above in the paper.

Lastly, we will **discuss Bell et al. (2023)'s [5] work** more extensively, which successfully applies the vanilla EPK formulation (equivalent to ExPLAIND scores accumulated on the data axis for standard gradient descent) for outlier detection and adversarial sample detection (i.e. other forms of data attribution) thus **further validating the data perspective of ExPLAIND.**

**References:**

\[1\] Schioppa, Andrea, Katja Filippova, Ivan Titov, and Polina Zablotskaia. “Theoretical and Practical Perspectives on What Influence Functions Do.” arXiv:2305.16971. Preprint, arXiv, May 26, 2023\. [https://doi.org/10.48550/arXiv.2305.16971](https://doi.org/10.48550/arXiv.2305.16971).

\[2\] Park, Sung Min, Kristian Georgiev, Andrew Ilyas, Guillaume Leclerc, and Aleksander Madry. “TRAK: Attributing Model Behavior at Scale.” arXiv:2303.14186. Preprint, arXiv, April 3, 2023\. [https://doi.org/10.48550/arXiv.2303.14186](https://doi.org/10.48550/arXiv.2303.14186).

\[3\] Pruthi, Garima, Frederick Liu, Mukund Sundararajan, and Satyen Kale. “Estimating Training Data Influence by Tracing Gradient Descent.” arXiv:2002.08484. Preprint, arXiv, November 14, 2020\. [https://doi.org/10.48550/arXiv.2002.08484](https://doi.org/10.48550/arXiv.2002.08484).

\[4\] Koh, Pang Wei, and Percy Liang. “Understanding Black-Box Predictions via Influence Functions.” *Proceedings of the 34th International Conference on Machine Learning*, PMLR, July 17, 2017, 1885–94. [https://proceedings.mlr.press/v70/koh17a.html](https://proceedings.mlr.press/v70/koh17a.html).

[5] Bell, Brian Wesley, Michael Geyer, David Glickenstein, Amanda S. Fernandez, and Juston Moore. “An Exact Kernel Equivalence for Finite Classification Models.” Paper presented at TAG-ML. January 1, 2023. https://openreview.net/forum?id=XygOJZa6QK.

---

### Author Response · Authors · 2025-12-04

We thank the reviewers and AC for their time and effort. We are happy to see that all reviewers agree on the novelty and impact of our theoretical extensions of the EPK, our unified explainability framework, and our experimental validation and application of it. Despite the very positive impressions in the strengths sections of the reviews, some weaknesses were raised. Sadly, no discussion was possible before the communication ban due to the OpenReview incident, and therefore some open points (and our extensive rebuttal to them) are not considered in the current state of the reviews.

In addition to addressing all the minor points regarding clarity and the discussion of our results (see rebuttals below), we wish to emphasize that we have addressed the following core weaknesses raised by several reviewers:

1. **Validation of data attribution.** As we state in the general comment below, **our submission now includes a validation experiment** in line with other works on evaluating data attribution, showing that ExPLAIND is performant in such scenarios. Furthermore, we now provide additional discussion comparing the data attribution provided by ExPLAIND vs. previous approaches (model-specific attribution vs. leave-one-out-and-retrain, see below). Finally, Bell et al. (2023)'s work is further evidence of the applicability of ExPLAIND for data attribution, which we also now discuss in our work.

2. **Large scale scenarios.** As we highlight in Section 4.1, simple computational tricks make ExPLAIND applicable to large-scale scenarios as well. To demonstrate this empirically, **we now include a GPT-2 model (100M parameters)** in the decomposition validation experiments, effectively showing that ExPLAIND can be applied to such large scales.

3. **Discussion of Grokking results.** While our results on Grokking provide intriguing insights into the training dynamics of the modulo-arithmetic Transformer model, our results do not show to what extent these insights generalize to other scenarios. As our main goal for this paper was to introduce and validate a novel, unified approach to interpretability, we consider such further studies out of scope for this submission. Addressing the concerns of the reviewers, we have made the scope of our results more explicit and extended *Future Work* stating clear directions for such a more general insight into these dynamics.

Thank you again for your consideration.

---

### Meta-Review · Area_Chair_uMPF · 2026-01-07

**Summary:**

Reviewers agree that the paper presents a technically solid and novel theoretical extension of Exact Path Kernel methods to realistic optimizers (e.g., AdamW), and that the goal of unifying model, data, and training-step attribution is conceptually appealing. However, the recommendation is driven by concerns that the empirical evidence and practical validation do not sufficiently support the scope of the claims. In particular, the evaluation relies heavily on toy or small-scale settings, offers limited quantitative validation beyond parameter pruning, and leaves open questions about scalability, usability, and general applicability of the proposed framework.

**Reviewer Concerns:**

Concerns addressed by the rebuttal:

The authors provide additional discussion and experiments to partially validate data attribution, including comparisons to existing influence-style methods.

A new GPT-2 (100M parameter) reconstruction experiment is introduced to suggest that parts of the framework can be applied at larger scales.

Several technical and clarity issues raised by reviewers (e.g., optimizer formulation, mini-batch attribution, interpretation of figures) are addressed.

Outstanding concerns:

Practical scalability and applicability remain unclear: ExPLAIND still requires access to full training trajectories and incurs high memory and computational overhead, limiting applicability to many real-world and pretrained models.

Empirical validation is narrow: quantitative evaluation focuses primarily on parameter pruning, with data- and training-step attribution remaining weakly validated and lacking standardized benchmarks.

Interpretability methodology lacks guidance: the framework yields a high-dimensional influence tensor, but there is no principled method for selecting aggregation axes or granularity, making analyses difficult to reproduce and conclusions potentially subjective.

Generality of insights is uncertain: the grokking case study is informative but largely qualitative and task-specific, with limited evidence that the findings or methodology generalize beyond controlled settings.

**Reviewer Scores:**

Reviewer 6LT3 (Reject): Likely unchanged. Core concerns about scalability, reliance on training trajectories, and limited validation remain unresolved.

Reviewer F7oz (Reject): Possibly slightly improved but still below acceptance; concerns about usefulness, scope, and experimental realism largely persist.

Reviewer ZFtG (Accept): Likely unchanged and positive.

Reviewer Sg8X (Borderline Reject): Likely modest improvement, but remaining concerns prevent a clear shift to accept.

---

### Decision · Program_Chairs · 2026-01-26

Reject